# Building bridges through dynamic coupling for organic phosphorescence

Xin Li [1,5], Wenlang Li[1,5], Ziqi Deng[2], Jingtian Wang[3], Shan He[1], Xinwen Ou [1], David Lee Phillips [2], Guanjun Xiao [3] ✉, Bo Zou [3], Ryan T. K. Kwok [1], Jianwei Sun [1], Jacky W. Y. Lam [1] ✉, Zhihong Guo [1] ✉ & Ben Zhong Tang [1,4] ✉

Achieving long-lived room temperature phosphorescence (RTP) in organic materials has garnered significant attention in the field of optoelectronics. Although many host–guest systems with versatile performances have been developed, their photophysical mechanisms remain unclear due to the complicated intermolecular interactions and multiple energy transfer pathways, leading to unavoidable trial-and-error in molecular designs. Here we reveal that the dynamic coupling process in the excited state is crucial for inducing phosphorescence, where host and guest molecules firstly couple to enhance the intersystem crossing efficiency, and then decouple to transfer excitons to the triplet state of guest. Such a process shows universal applicability and tunable performance, with the longest lifetime for red RTP ($\tau_P$ = 2.4 s) reported so far. We anticipate the present work as a starting point for more sophisticated models on excited-state dynamic behaviors within host–guest systems.

The phosphorescence emitted from the triplet state of chromophores exhibits a long lifetime ranging from microseconds to seconds, making it promising for applications in optoelectronics[1–3], photocatalysis[4,5], and biomedicals[6–9]. Recently, the development of pure organic phosphorescent materials has become a hot research topic due to their superior flexibility, processability and biocompatibility[10–18]. But their performances are constrained by inefficient intersystem crossing (ISC) and intense non-radiative decay at room temperature. Strategies such as incorporation of heavy atoms[19–21] or ($n$, $\pi^*$) transitions[12,22,23] to accelerate ISC, crystallization[13,24] or deuteration[25] to suppress non-radiative pathway, are developed to enable room temperature phosphorescence (RTP) in single-component systems. However, such systems require intricate molecular design and crystal engineering, and often suffer from self-quenching[11,14]. Therefore, host–guest binary systems, where guest molecules are dispersed in host matrices, stand

out due to their facile fabrications, tunable emissions and enhanced performances[26–30].

In the early stage, the hosts primarily functioned as a rigid matrix and oxygen barrier to stabilize the triplet excitons of the guests[11,26,31,32]. The phosphorescence performance relied on the ISC efficiency of guests, making it challenging to induce phosphorescence of guests with poor inherent ISC process. Later, energy transfer (ET) process was introduced[33–36], where the host with high ISC efficiency acted as a reservoir for triplet excitons and transferred them to the guest, thereby expanding the scope of guest candidates. In this classical framework, the excitation energy was localized in either host or guest molecules, with only excitons hopping between these discrete localized states. However, there existed complicated intermolecular interactions spanning across molecules at varying proximities such as Coulomb interactions[37], electron exchange[38], quantum coherence[39],

[1]Department of Chemistry and the Hong Kong Branch of Chinese National Engineering Research Center for Tissue Restoration and Reconstruction, The Hong Kong University of Science and Technology, Clear Water Bay, Kowloon, Hong Kong 999077, China. [2]Department of Chemistry, The University of Hong Kong, Pokfulam, Hong Kong 999077, China. [3]State Key Laboratory of Superhard Materials, College of Physics, Jilin University, Changchun 130012, China. [4]Guangdong Basic Research Center of Excellence for Aggregate Science, School of Science and Engineering, The Chinese University of Hong Kong, Shenzhen (CUHK-Shenzhen), Shenzhen 518172, China. [5]These authors contributed equally: Xin Li, Wenlang Li. ✉e-mail: xguanjun@jlu.edu.cn; chjacky@ust.hk; chguo@ust.hk; tangbenz@cuhk.edu.cn

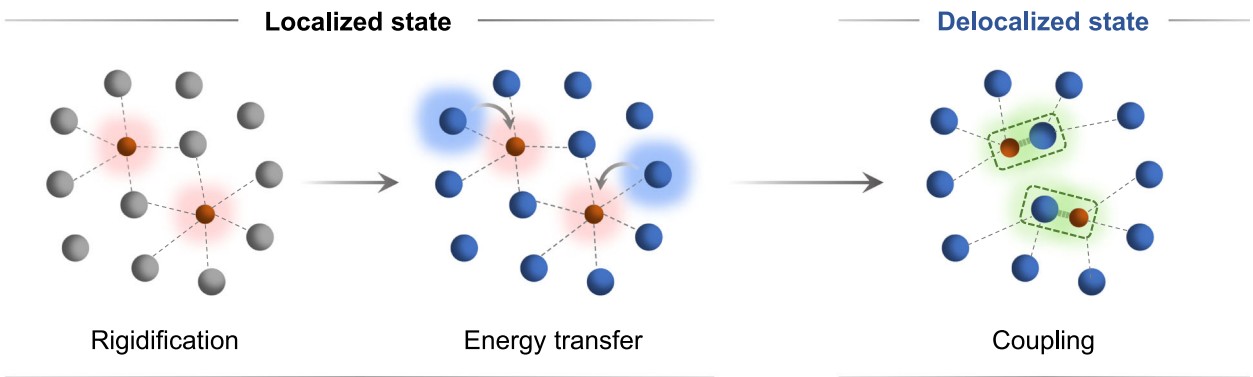

**a Historical development: RTP mechanism in host–guest systems**

Localized state · Delocalized state

Rigidification   Energy transfer   Coupling

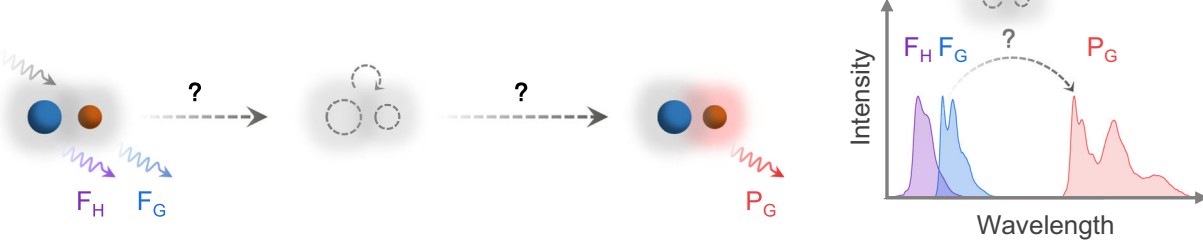

**b Previous work: Static model involving a 'ghost' intermediate**

$F_H$  $F_G$   $P_G$

Intensity — Wavelength

$F_H$ $F_G$   ?   $P_G$

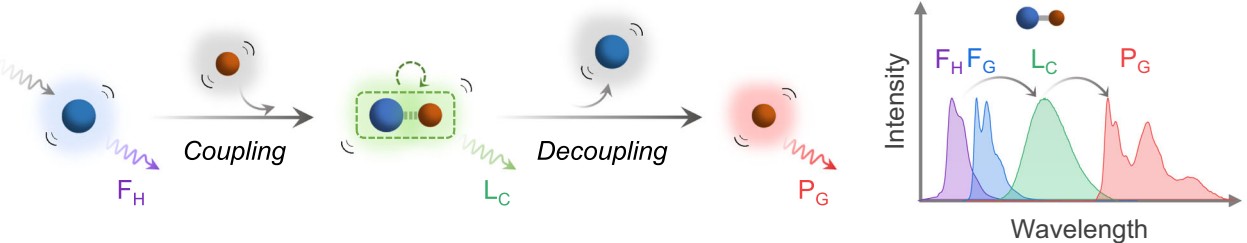

**c This work: Dynamic coupling model involving a 'visible' complex**

$F_H$   Coupling   $L_C$   Decoupling   $P_G$

Intensity — Wavelength

$F_H$ $F_G$   $L_C$   $P_G$

**Fig. 1 | Scheme representation of dynamic coupling in host–guest RTP systems. a** Historical development of RTP mechanism in host–guest systems. **b** Static model and spectral characteristics in previous RTP work. Either host or guest molecules (or both) were firstly excited to the singlet state, followed by possible transfer of excitons through a 'ghost' intermediate state to the triplet state of guest, activating RTP of guest emitter. Only fluorescence of host and guest, as well as phosphorescence of guest were observed in the spectra. All species, including host, guest and 'ghost' complex, were static and remained unchanged throughout the photophysical process. **c** Dynamic coupling model and spectral characteristics in this work. Upon excitation of the host molecules, the collision between host and guest molecules formed a spectroscopically 'visible' complex, which subsequently dissociated to activate the RTP of guest. Such a dynamic complex with enhanced intersystem crossing (ISC) ability worked as a bridge to transfer excitons from the singlet state of host to the triplet state of guest and could be observed in the spectra. $F_H$, fluorescence of host; $F_G$, fluorescence of guest; $L_C$, emission of complex; $P_G$, phosphorescence of guest. Blue spheres represented host molecules, and red for guest molecules. Gray arrows represented UV excitation. The dotted curved arrows represented ISC process of complex.

etc. These interactions could couple neighboring host and guest molecules together to form a complex with excitation energy delocalized among them (Fig. 1a)[40–43].

Surprisingly, the complexes were less investigated or just vaguely mentioned in previous RTP studies, the difficulty of such research was due to the lack of experimental evidence, unclear understanding of its coupling interaction, and ongoing debate regarding its impact on RTP (facilitating or hindering)[44–47]. Additionally, the host, guest, and complex were always perceived as being frozen in their respective positions throughout the photophysical process. This static model failed to capture the dynamic nature within the host–guest system, where the continuous molecular motions would affect the intermolecular orbital overlap and reorganize the distribution of excitation energy[48–50]. Such

ambiguous and oversimplified mechanism necessitated unavoidable trial-and-error in the design of host–guest RTP system (Fig. 1b).

Therefore, this work proposed a dynamic coupling model to further understand the photophysical process of RTP in host–guest systems. The host and guest molecules were non-covalently coupled in the excited state through intermolecular charge-transfer (CT). Thus, the coupling or decoupling process of the complex could be delicately influenced by the intermolecular motions (collision or dissociation) (Fig. 1c). The spectroscopic evidence confirmed the existence of the complex and verified the underlying photophysical mechanism for facilitating RTP. For the first time we clarified that such dynamic coupling was essential for RTP of guest emitter, where static coupling failed to do so. The universality of this dynamic coupling-induced RTP

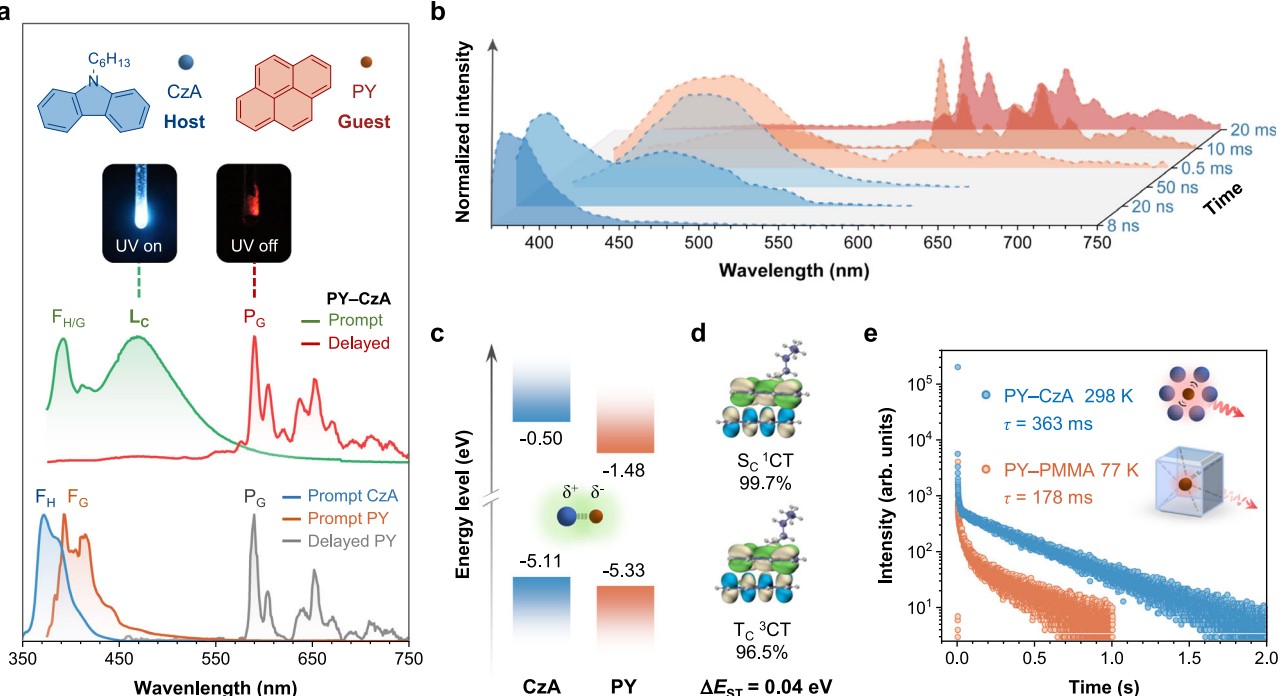

**Fig. 2 | Observation of coupling in host−guest RTP system. a** Prompt and delayed photoluminescence (PL) spectra of PY−CzA under ambient conditions excited at 365 nm. The prompt PL spectra of CzA crystals and PY−PMMA film, as well as the delayed PL spectra of PY−PMMA film at 77 K were included for comparison. Delay time: 10 ms. Chemical structures of CzA and PY were illustrated. The inset showed the photographs of PY−CzA crystals taken under 365 nm UV on and off modes. **b** Time-resolved PL spectra of PY−CzA at varying delay time. **c** Energy level alignment of frontier molecular orbitals of CzA and PY. **d** Natural transition orbitals (NTOs) contributing to the transitions for singlet and triplet states of the complex ($S_C$ and $T_C$). The energy difference between $S_C$ and $T_C$ ($\Delta E_{ST}$) was noted. Color code: green, hole; blue, particle. **e** Lifetime curves of PY−CzA crystals at 298 K and PY−PMMA film at 77 K excited at 365 nm with emission recorded at 595 nm.

was demonstrated in various systems with multicolor phosphorescence obtained. Furthermore, such a dynamic process could be further regulated through molecular design to enhance performance. By combining the deuteration strategy, red RTP with an unprecedented ultralong lifetime of 2.4 s in air was achieved, representing the longest red RTP observed so far. Moreover, we showcased the potential applications of these host−guest systems in four-dimensional (4D) encryption and temperature-dependent full-color display.

## Results

### Observation of host−guest coupling

A simple host−guest system called PY−CzA was fabricated, with *N*-hexyl carbazole (CzA) as host and pyrene (PY) as guest through melt-casting at a doping ratio of 1.0 wt%. CzA was synthesized in lab and well characterized (Supplementary Section II) since the impurities in commercial carbazole derivatives were reported to significantly impact their photophysical properties[29]. PY−CzA exhibited a bright sky-blue emission under 365 nm UV irradiation and red RTP after ceasing the excitation source, while neither individual CzA nor PY displayed red RTP phenomenon (Fig. 2a and Supplementary Figs. 31 and 32). The consistent powder X-ray diffraction patterns of PY−CzA and CzA indicated the same molecular packing before and after doping, excluding the effect of crystal-phase change (Supplementary Fig. 33). Notably, besides the fluorescence of Cz and PY ($F_{H/G}$), a new peak was observed at 470 nm in the prompt photoluminescence (PL) spectrum of PY−CzA, which was assigned to the emission of the complex ($L_C$)[43]. This spectroscopically 'visible' complex verified the occurrence of coupling in PY−CzA. The delayed spectrum matched well with the phosphorescence spectrum of PY dispersed in poly (methyl methacrylate) (PY−PMMA) at 77 K, indicating the red RTP originated from the phosphorescence of PY ($P_G$). The lack of RTP in CzA and PY themselves but its appearance in PY

−CzA with $L_C$ suggested a potential correlation between coupling and RTP. Interestingly, the time-resolved PL spectra revealed a gradual transition from $F_{H/G}$ to $L_C$ in the nanosecond range and then to $P_G$ in the millisecond range (Fig. 2b), indicating that: (1) $L_C$ at 470 nm was a mixed emission from both short-lived singlet and long-lived triplet states of complex ($S_C$, $T_C$) (Supplementary Fig. 34) and (2) the complex may serve as an intermediate state to facilitate the exciton transfer from the singlet states of Cz or PY ($S_{H/G}$) to the triplet state of PY ($T_G$). To study the nature of coupling interaction in the complex, Fig. 2c illustrated that the highest-occupied molecular orbital (HOMO) of CzA was higher than that of PY whereas the lowest-unoccupied molecular orbital (LUMO) of PY was lower. This demonstrated a typical energy level alignment allowing for inter-molecular CT process[40]. The coupling of complex through inter-molecular CT was further supported by natural transition orbitals (NTOs) contributing to the transitions for $S_C$ and $T_C$ states. The hole was distributed on CzA (donor) while particle was distributed on PY (acceptor), which was consistent with their HOMO and LUMO alignments (Fig. 2d). Such a CT characteristic provided a nearly degenerate $S_C$ and $T_C$ with a negligible energy gap ($\Delta E_{ST}$) of 0.04 eV, indicating an efficient ISC process in the complex. More surprisingly, PY−CzA exhibited a phosphorescence lifetime of 363 ms at 298 K, which was more than 2-fold the value of PY−PMMA at 77 K (178 ms) under nitrogen atmosphere (Fig. 2e). Despite the occurrence of more non-radiative decay at 298 K compared to PY−PMMA under cryogenic conditions, PY−CzA exhibited a brighter and longer phosphorescence. This counterintuitive phenomenon suggested the presence of additional intermolecular exciton transfer pathways that enhanced the exciton population on $T_G$ in PY−CzA, driving us to investigate the underlying photophysical process and elucidate the relationship between intermolecular coupling and the generation of RTP.

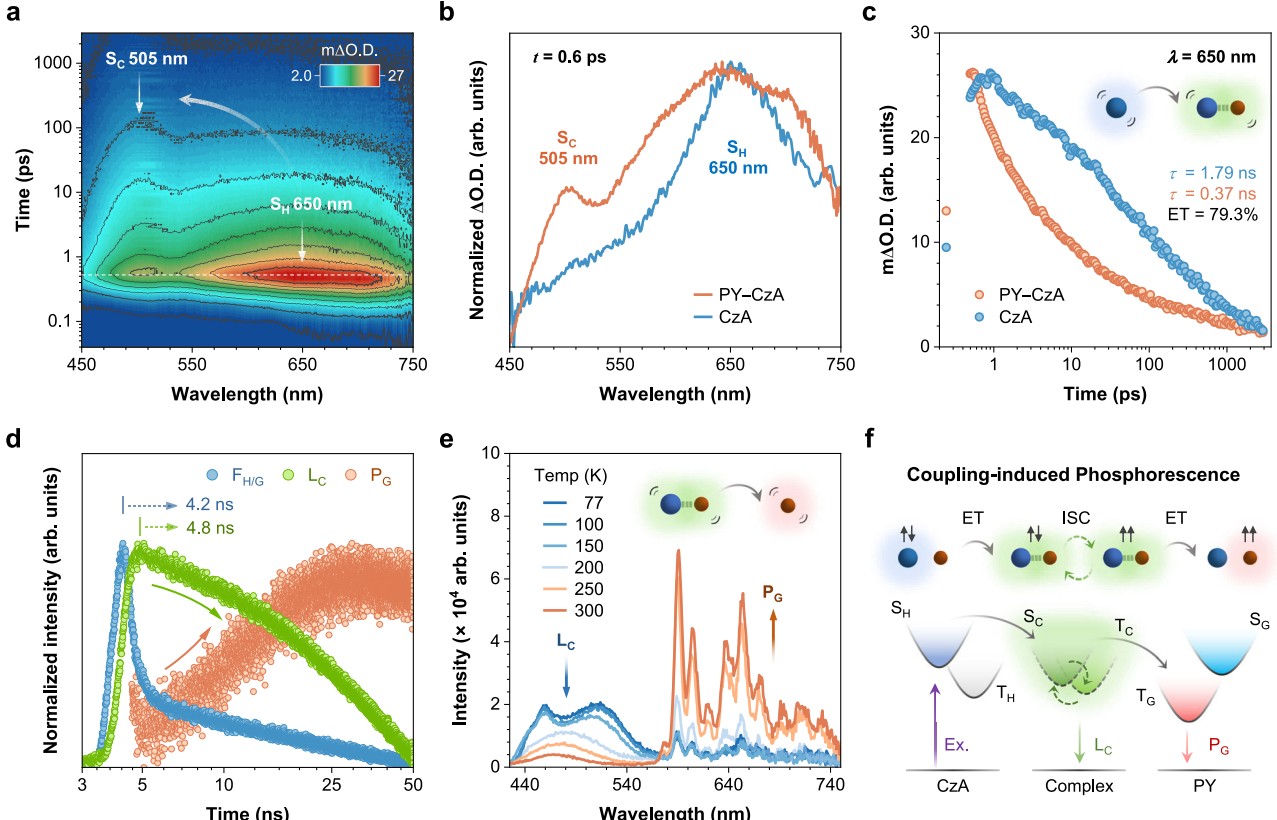

**Fig. 3 | Photophysical mechanism investigation. a** Three-dimensional (3D) contour map of fs-TA spectra for PY−CzA crystals measured at 267 nm excitation wavelength. Absorption of the singlet states of CzA and complex ($S_H$ and $S_C$). **b** TA spectra of PY−CzA crystals and pure CzA crystals at a pump-probe delay time of 0.6 ps. **c** Absorption kinetics of $S_H$ in PY−CzA crystals and pure CzA crystals recorded at 650 nm. Fitted lifetime values and energy transfer efficiency were indicated. **d** Lifetime curves of emission from different species in PY−CzA crystals within a 50 ns range. Emission wavelength recorded for $F_{H/G}$, 400 nm; $L_C$, 470 nm; $P_G$, 595 nm. The lifetime curve of $P_G$ generation was obtained through subtracting the $L_C$ component from the lifetime curve recorded at 595 nm. The time points at which the intensity of $F_{H/G}$ and $L_C$ reached their maximum were noted. **e** Temperature-dependent delayed PL spectra of PY−CzA at a delay time of 10 ms excited at 365 nm. **f** The proposed photophysical process of coupling-induced phosphorescence in PY−CzA system. ET, energy transfer; ISC, intersystem crossing. Blue spheres represented host molecules, and red for guest molecules. The gray arrows represented ET process, and green dotted arrows represented ISC process of complex.

## Coupling-induced phosphorescence

Femtosecond transient absorption (fs-TA) spectroscopy was first performed on PY−CzA crystals to capture the transient species in the excited state within 3 ns. Upon excitation, an excited-state absorption peak at 650 nm promptly emerged, which subsequently shifted to 505 nm within 100 ps (Fig. 3a). By comparing with fs-TA spectra of the host and guest (Fig. 3b, Supplementary Figs. 35 and 36), the initially generated peak was assigned to the singlet state of host ($S_H$), indicating CzA was firstly excited. After 100 ps, the newly aroused peak at 505 nm dominated, which was absent in either CzA or PY. Thus, it can be assigned to the absorption of $S_C$. By monitoring the absorption kinetics of $S_H$, a significant decrease in lifetime was observed, from 1.79 ns in pure CzA crystals to 0.37 ns in PY−CzA crystals (Fig. 3c and Supplementary Fig. 37). This indicated the transfer of excitons from $S_H$ to $S_C$ during coupling process and the ET efficiency could be quantitively determined to be 79.3%.

To probe how the transient species evolved after $S_C$, we extended the time range to 50 ns and measured the emission kinetics of different species. Figure 3d showed $L_C$ aroused and reached its maximum intensity at time later than $F_{H/G}$ by 0.6 ns. This supported the coupling process occurred after CzA was excited. Later, $L_C$ started to decay, while $P_G$ simultaneously began to increase and reached the maximum intensity at around 25 ns, suggesting the population of excitons on $T_G$ possibly originated from the complex (Supplementary Fig. 38).

Moreover, a gradual emission shift from $L_C$ to $P_G$ within 1 ms was also observed through time-resolved emission spectroscopy (TRES) (Supplementary Fig. 39). Considering the small $\Delta E_{ST}$ of the complex, a rapid ISC process from $S_C$ to $T_C$, followed by an ET process from $T_C$ to $T_G$, was proposed.

To further prove the occurrence of ET process, temperature-dependent delayed PL spectra were measured (Fig. 3e). At low temperature, PY−CzA exhibited a mixture of phosphorescence of host crystals, $P_H$, and complex emission, $L_C$, in the 420-540 nm range, along with a weak phosphorescence of guest molecules in the 590-750 nm range. As the temperature increased to 200 K, $P_H$ disappeared due to thermal quenching and $L_C$ became distinctly observable. At the same time, the intensity of phosphorescence of guest, $P_G$, started to increase, indicating that the presence of complex favored the formation of phosphorescence. As the temperature continued to rise, instead of being weakened by enhanced thermal vibration, $P_G$ significantly intensified while $L_C$ diminished, confirming the existence of a thermally activated ET process from $T_C$ to $T_G$ (Supplementary Fig. 40). Therefore, the photophysical process involved in the coupling-induced phosphorescence was summarized in Fig. 3f. The excitons were firstly excited to $S_H$ and transferred to $S_C$ with high efficiency during coupling process. The complex then underwent a fast ISC process and served as a bridge to populate more excitons to $T_G$, thus enhancing $P_G$ performance of PY−CzA as discussed in Fig. 2e.

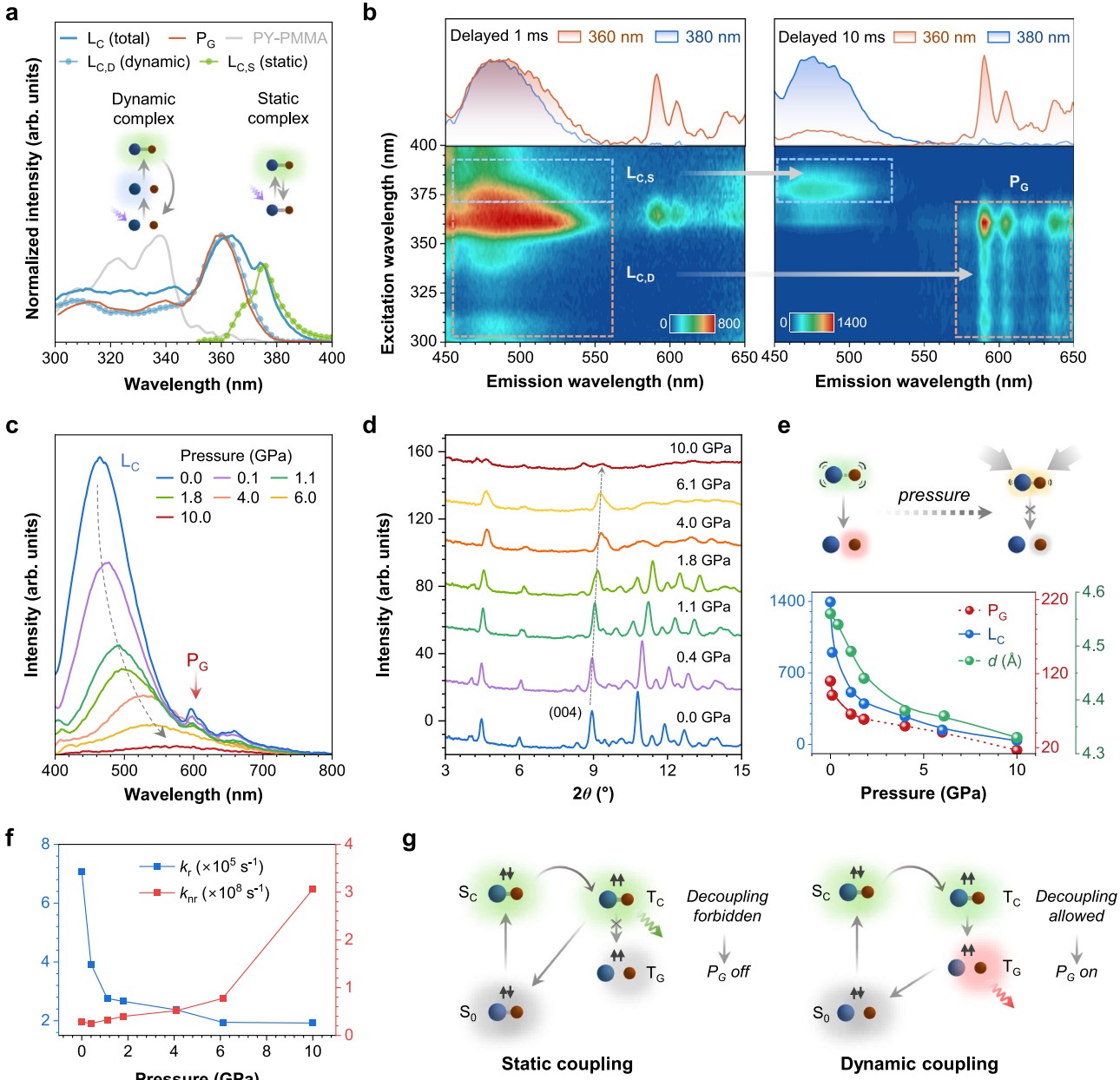

**Fig. 4 | Demonstration for necessity of dynamic coupling. a** Analysis of excitation spectra of PY−CzA. The excitation spectrum of the complex emission was measured at 470 nm ($L_C$; blue solid line). The excitation spectrum of phosphorescence was measured at 595 nm with a delay of 10 ms ($P_G$; red solid line). The excitation spectrum of the dynamic complex component was obtained by measuring the excitation spectrum of CzA crystals ($L_{C,D}$; line with blue circles). The excitation spectrum of the static complex component was obtained by subtracting the dynamic component from the excitation spectrum of the total complex emission ($L_{C,S}$; line with green circles). The excitation spectrum of PY−PMMA was attached for reference (gray line). Inset: the difference between dynamic and static complexes. **b** Excitation-phosphorescence mapping of PY−CzA with a delay of 1 (left) and 10 ms (right). The upper insets: the delayed PL spectra excited at 360 nm and 380 nm. $P_G$ appeared through dynamic coupling when PY−CzA was excited at 360 nm but disappeared when static complex was excited at 380 nm. **c** Pressure-dependent PL spectra of PY−BrCzA excited at 355 nm. **d** Pressure-dependent X-ray diffraction patterns of PY−BrCzA crystals. The diffraction peak of (004) crystal plane was denoted. **e** The variations in the intensities of $L_C$ and $P_G$, along with the interplanar spacing of (004) crystal plane, as external pressure increased. **f** Rate constants of radiative and non-radiative decay from $S_C$ state within PY−BrCzA complex under different pressure. **g** Schematic diagram for different impact of static (left) and dynamic (right) coupling on RTP ($P_G$). In static coupling, the static complex was already formed in the ground state. Upon excitation to $S_C$ and subsequent ISC to $T_C$, the static complex remained coupled and decayed back to the ground state, exhibiting $L_C$ emission. In dynamic coupling, the dynamic complex was formed after excitation to the excited state. After undergoing ISC process to $T_C$, the dynamic complex dissociated and facilitated the transfer of excitons to $T_G$. Thus, the excitons decayed from $T_G$, exhibiting $P_G$ emission.

## The necessity of dynamic coupling

To further investigate the excited-state behaviors of complex involved in this coupling-induced phosphorescence, excitation spectrum of PY−CzA was measured at $L_C$ (Fig. 4a). It was different from the excitation spectra of PY−PMMA film and PY crystals, excluding the possibility that

$L_C$ originated from guest molecules (Supplementary Fig. 41). From 300 nm to 373 nm, the spectrum resembled that of CzA crystals, indicating an exciplex formed through coupling excited-state CzA and ground-state PY. However, a new shoulder peak appeared at approximately 375 nm, coinciding with a newly observed absorption

peak in the UV-Vis absorption spectrum (Supplementary Fig. 42). This suggested the formation of a ground-state absorption complex. The exciplex coupled only in the excited state but dissociated in the ground state, while the absorption complex coupled in both the excited and ground states, hence referred to as dynamic and static complex. Thus, the excitation spectrum recorded at $L_C$ (total) could be deconvolved into two components: excitation of the dynamic complex ($L_{C,D}$) and the static complex ($L_{C,S}$). Interestingly, the excitation spectrum recorded at $P_G$ closely matched the dynamic component, but distinctly different from the static component. This indicated that the excitons of $T_G$ and dynamic complex shared the same excitation pathway, while differing from the static complex.

To monitor the evolution of dynamic and static complex over time, excitation-phosphorescence mapping was performed on PY−CzA with a delay time of 1 and 10 ms, respectively (Fig. 4b). At 1 ms delay, the complex emission dominated from 450 to 550 nm, showing an excitation pattern composed of dynamic and static components ($L_{C,D}$ and $L_{C,S}$) as discussed earlier. $P_G$ was weak and exclusively located within the excitation range of $L_{C,D}$. As the delay time prolonged to 10 ms, the $L_{C,D}$ region disappeared, while the $P_G$ region enhanced and exhibited an excitation pattern parallel to $L_{C,D}$ at 1 ms delay. This indicated a gradual transition from $L_{C,D}$ to $P_G$. However, the emission of static component remained in its original region, exhibiting only decay behavior. This suggested that despite the degenerate energy levels of dynamic and static complexes (indistinguishable emission spectra as shown in the upper inset), only excitons from the dynamic complex could transfer to $T_G$, while excitons from the static complex were trapped and decayed at $T_C$.

Since the difference between dynamic and static complexes lay in whether they remained coupled during the decay to the ground state, it could be speculated that a similar dissociation process might be involved in the excited-state transition from $T_C$ to $T_G$ (Supplementary Figs. 43 and 44). To verify this, pressure-dependent experiments were conducted to regulate the coupling strength. PY−BrCzA was prepared by doping PY into brominated CzA to enhance the $P_G$ intensity through heavy atom effect, enabling the simultaneous observation of $L_C$ and $P_G$ (Supplementary Figs. 45 and 46). In situ pressure-dependent PL spectra (Fig. 4c) shows the emission ($L_C$) gradually red-shifted with weaker intensity as the pressure increased. This clearly demonstrated the progressively enhanced CT character of complex state, indicating that the intermolecular coupling was strengthened. However, contrary to the anticipated enhancement due to the suppression of molecular vibrations under high pressure[51], $P_G$ weakened and eventually disappeared. This abnormal phenomenon might be because the decoupling of complex was also constrained by the reduction in the free molecular volume under high pressure.

More structural characterizations and quantitative analysis were carried out. X-ray diffraction showed that all diffraction peaks shift toward higher angles with increasing pressure, indicating significant reduction in unit cell volume and interplanar spacing (Fig. 4d and Supplementary Fig. 47). Taking (004) crystal plane as an example, we calculated its interplanar spacing, which showed a continuous decreasing from 4.56 Å to 4.33 Å. Figure 4e displayed the variation in the intensities of $L_C$ and $P_G$, and (004) interplanar spacing under different pressure. Surprisingly, the trend of weakening intensities of $L_C$ and $P_G$ was highly consistent with the trend of reduced interplanar spacing (Supplementary Figs. 48 and 49). Quantitative analysis further investigated the relationship between activation energy during the decoupling process and RTP performance under various pressure (Supplementary Fig. 50). A robust linear correlation ($R^2 = 0.99$) provided compelling experimental validation for our proposed mechanism: shortening the intermolecular distance within complex enhanced the coupling strength and elevated the decoupling activation energy, thereby diminishing the RTP performance.

Furthermore, we observed a continuous bathochromic shift in UV-Vis absorption spectra and progressively shortened lifetimes of $L_C$ with increasing pressure (Supplementary Figs. 51 and 52), enabling calculation of the radiative ($k_r$) and non-radiative ($k_{nr}$) decay rate constants (Fig. 4f and Supplementary Table 5). The decrease in $k_r$ originated from diminished HOMO-LUMO orbital overlap resulting from enhanced CT character, while the increase in $k_{nr}$ was due to that the shortened intermolecular distance during the pressurization process intensified collisions and nonradiative energy transfer. Complementary FT-IR and Raman spectra further confirmed the pressure-enhanced intermolecular interactions (Supplementary Figs. 53−55). Together, these experiments demonstrate how external pressure can precisely modulate the coupling strength within the PY−BrCzA system, providing compelling evidence that validated the necessity of dynamic coupling in generating RTP through *reductio ad absurdum*.

Thus, the different impacts of static and dynamic coupling on RTP were summarized in Fig. 4e. For the static complex in the excited state, rather than undergoing significant conformational reconfiguration to a dissociated $T_G$ state, it preferred decaying directly to the coupled ground state. While for the dynamic complex which would eventually dissociate at ground state, the decoupling process was unlocked to allow exciton transfer to $T_G$ to activate $P_G$.

## Structure−property relationship for dynamic coupling

To enrich the emission color and expand the applicable scope of dynamic coupling-induced RTP, 17 systems were screened by using CzA/BrCzA and fluorene derivatives as hosts, along with 8 guest molecules possessing various triplet energy levels (Supplementary Fig. 56). Notably, multicolor RTP was achieved, highlighting the universality of this approach (Supplementary Fig. 57). Four of them exhibiting RTP in green, yellow, red, and near-infrared regions were presented as representatives (Fig. 5a, b). By comparing with the phosphorescence spectra of guest molecules, it was confirmed that these RTP originated from the triplet states of the guests (Supplementary Fig. 58). Interestingly, dynamic coupling was observed in all systems exhibiting RTP, while systems that failed to induce RTP either lacked coupling or trapped excitons within the complex (Supplementary 59−71). This emphasized the indispensability of dynamic coupling for facilitating RTP.

Further analysis revealed that the different performances of these host−guest systems could be categorized into three scenarios, through which the design principle for host−guest RTP systems could be clarified (Supplementary Figs. 72 and 73). Considering that the coupling process required close proximity and matching orbital energy levels, while the decoupling process necessitated appropriate intermolecular interactions, the corresponding structure-property relationship for dynamic coupling-induced phosphorescence was deduced as follows: (1) Host and guest molecules should have a certain degree of planarity to enable close proximity to one another; (2) Their electron-donating and accepting tendencies should align within a specific range to form suitably attractive intermolecular charge transfer interactions without being too strong to trap the excitons. Based on this, we further showcased how to utilize such structure-property relationship to guide the design of host−guest RTP systems, and successfully activated RTP through regulating the coupling process in host−guest systems (Supplementary Fig. 74).

## Tuning dynamic coupling to enhance RTP performance

The term 'dynamic' implied the coupling process can be further tuned to improve the RTP performance. Thus, we modified CzA with benzoyl group and obtained two new host molecules called CzBP and BPCzA (Fig. 5c). Similar to CzA, they exhibited negligible RTP in pure crystals but showed brighter and longer red RTP after doping with PY, visible to naked eyes even under ambient light (Supplementary Figs. 75 and 76). The new broad peaks appeared in the prompt spectra confirmed the

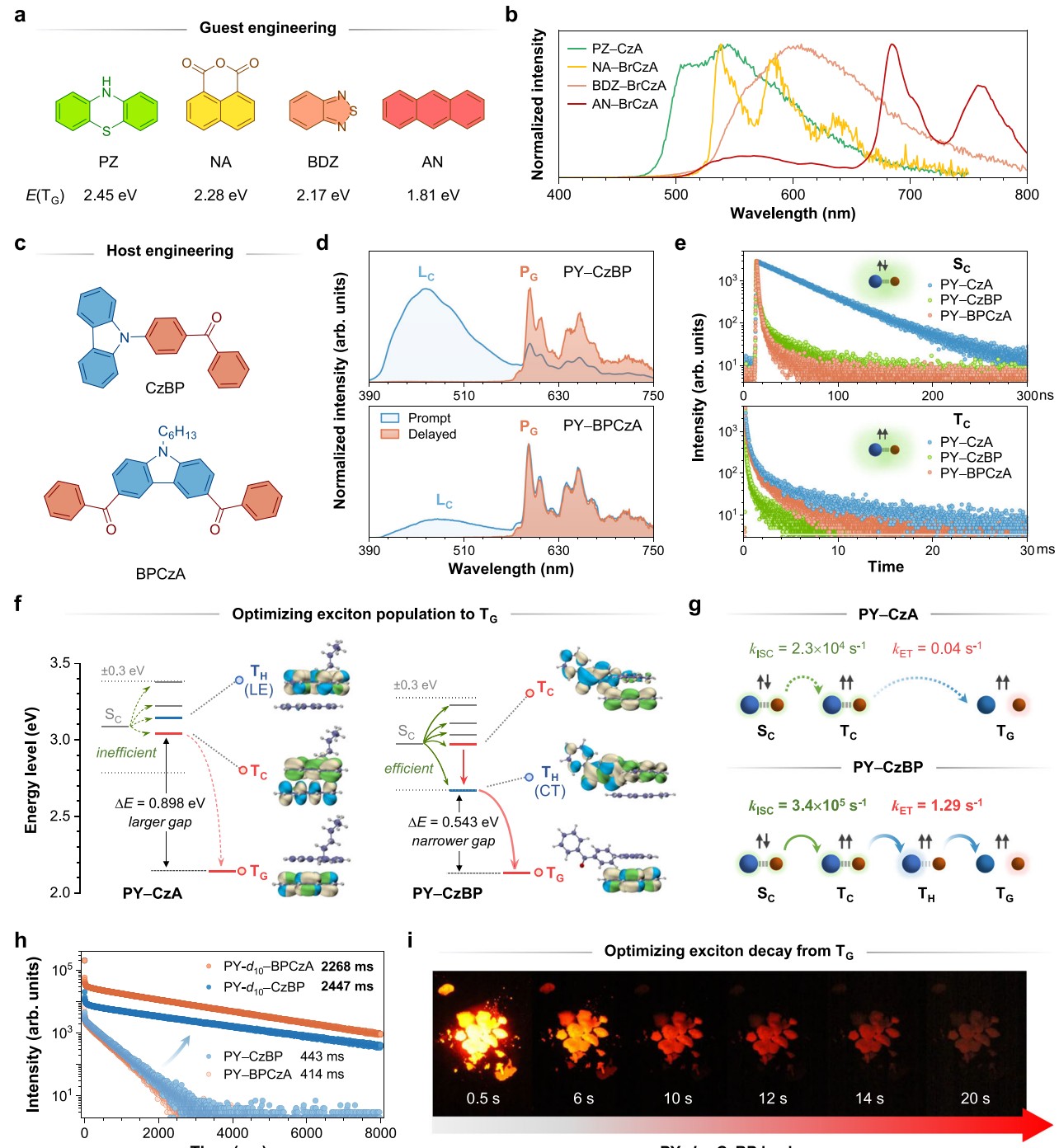

**Fig. 5 | RTP performance enhancement. a** Chemical structures of 4 representative guest molecules used in the universality study (PZ, NA, BDZ and AN). **b** Delayed PL spectra of 4 representative host–guest systems used in the universality study, excited at 365 nm. **c** Chemical structures of modified host molecules. **d** Prompt and delayed PL spectra of PY−CzBP (top) and PY−BPCzA system (bottom) excited at 365 nm. Delay time: 10 ms. The emission of complex ($L_C$) and phosphorescence of guest ($P_G$) were noted. **e** Lifetime curves of complex emission in the singlet state (top) and triplet state (bottom) in PY−CzA, PY−CzBP, and PY−BPCzA. Emission wavelength recorded: 470 nm for PY−CzA and PY−CzBP; 500 nm for PY−BPCzA. **f** Calculated energy levels of PY−CzA and PY−CzBP systems. Possible ISC channels were represented by green arrows and $\Delta E$ values for ET process were noted. Insets: NTOs contributing to $T_H$, $T_C$, and $T_G$ transitions in these systems. Color code: green,

hole; blue, particle. **g** The scheme representation showed that the intramolecular D-A structure of PY−CzBP/BPCzA systems accelerated the ISC process through providing more ISC channels and facilitated the decoupling process through lowering $T_H$ to serve as an additional bridge, to enhance the RTP performance. $k_{ISC}$, the experimental rate constant of ISC process; $k_{ET}$, the experimental rate constant of energy transfer process during decoupling. Blue spheres represented host molecules, and red for guest molecules. The green arrows represented the ISC process of complex and blue for ET process. **h** Lifetime curves recorded at $P_G$ in PY−CzBP/BPCzA, PY-$d_{10}$−CzBP/BPCzA, excited at 365 nm. The fitted lifetimes were noted. **i** Photographs of PY-$d_{10}$−CzBP crystals captured in air for 20 s after the 365 nm UV lamp was turned off.

occurrence of coupling in PY−CzBP and PY−BPCzA. The presence of structured $P_G$ bands observed in both prompt and delayed spectra indicated an enhancement in the RTP intensity compared to the PY−CzA system (Fig. 5d). DFT calculations revealed the complexes were coupled by CT interaction (Supplementary Fig. 77). TRES spectra, excitation spectra, UV-Vis absorption spectra, excited-state conformations, and NCI analysis confirmed their photophysical processes of dynamic coupling-induced RTP (Supplementary Figs. 78–84).

To investigate how the incorporation of benzoyl group affected the properties of complex and its subsequent influence on the RTP performance, the lifetime of complex emission in PY−CzA and PY−CzBP/BPCzA was compared (Fig. 5e and Supplementary Table 6). In nanosecond range, a significant decrease in the $S_C$ lifetime of PY−CzBP/BPCzA was observed, suggesting a potential enhanced ISC process. The calculated energy level diagram revealed that more triplet states were generated within the range of ±0.3 eV around $S_C$ to serve as ISC channels in PY−CzBP/BPCzA, due to their intramolecular donor-acceptor (D-A) structures (Fig. 5f and Supplementary Figs. 85–87). This result was consistent with their increased rate constants of ISC ($k_{ISC}$) (Supplementary Fig. 88). Meanwhile, the lifetime of $T_C$ decreased from PY−CzA to PY−BPCzA to PY−CzBP in the millisecond range, indicating an increasingly accelerated ET process from $T_C$ to $T_G$. DFT calculations showed a large energy gap $\Delta E(T_C\text{-}T_G)$ in PY−CzA hindered the ET process. Whereas in PY−CzBP/BPCzA, $T_H$ existed between $T_C$ and $T_G$, working as an intermediate energy level to facilitate the ET process with a smaller energy gap $\Delta E(T_H\text{-}T_G)$ (Fig. 5f and Supplementary Fig. 89). NTO analysis revealed that CT interaction within the intramolecular D-A structure in PY−CzBP/BPCzA lowered the energy of $T_H$, compared to pure locally-excited (LE) transition of $T_H$ observed in PY−CzA, thus providing this additional $T_H$ bridge. Indeed, the experimental rate constants for ET process, $k_{ET}$, were proportional to the reciprocal of energy gap (Supplementary Fig. 90).

Figure 5g illustrated that incorporating benzoyl group enhanced ISC efficiency of complex, as well as facilitated its decoupling, with significant increases in the rate constants of ISC and ET processes. Optimizing both processes enhanced the population of excitons to $T_G$, thereby extending the RTP lifetime to 414 ms and 433 ms and increasing the quantum yield ($\Phi_P$) by 10 times (Fig. 5h and Supplementary Fig. 91). To further optimize the decay process from $T_G$ to $S_0$, PY was deuterated to suppress the non-radiative decay (Supplementary Figs. 92–95). A lifetime of 2447 ms was achieved in PY-$d_{10}$−CzBP with red RTP lasting over 20 s (Fig. 5h,i). This represented the current world record for the longest red RTP.

Beyond qualitative observations, we established a semi-empirical physical parameter to quantitatively characterize the dynamic coupling interaction (Supplementary Section VII). The activation energy barrier for the decoupling process ($\Delta G_{dc}^{\ddagger}$) was calculated based on Marcus theory[52,53] and systematically compared across various host−guest systems (Supplementary Figs. 96–99, Supplementary Tables 7 and 8). Our analysis revealed a strong inverse correlation ($R^2 = 0.99$) between $\Delta G_{dc}^{\ddagger}$ and RTP performance: higher energy barriers consistently corresponded to diminished phosphorescence lifetime (Supplementary Fig. 98). These quantitative results not only validated decoupling dynamics as a reliable regulatory mechanism for phosphorescence, but also shedding light on the design principles for host−guest RTP systems.

Thus, after reasonably tuning the dynamic coupling process, RTP performance of PY, a guest with poor intrinsic ISC efficiency, was significantly enhanced, surpassing a series of well-established host−guest systems (Table 1). For instance, in PY−PMMA system, phosphorescence was not observable at room temperature. In the widely recognized amorphous host, β-estradiol[25], $\Phi_P$ was only 0.002%, accompanied by a short lifetime of 134 ms. However, in dynamic coupling system PY−BrCzA, $\Phi_P$ significantly increased to 5.9%, demonstrating a remarkable enhancement of 2950 times higher than β-estradiol system. Similarly,

**Table 1 | Comparison of RTP performance among different host−guest systems**

| Sample | System | $\Phi_P$ (%)[a] | $\tau_P$ (ms) |
|---|---|---|---|
| PY−PMMA | Polymer | /[b] | 178[c] |
| PY−β-estradiol | Amorphous steroid | 0.002 | 134 |
| PY−BrCzA | Dynamic coupling | 5.9 | 104 |
| PY−CzBP | | 0.8 | 433 |
| PY−BPCzA | | 1.1 | 414 |
| PY-$d_{10}$−PMMA | Polymer | 0.01 | 98 |
| PY-$d_{10}$−β-estradiol | Amorphous steroid | 0.02 | 1536 |
| PY-$d_{10}$−BrCzA | Dynamic coupling | 9.8 | 216 |
| PY-$d_{10}$−CzBP | | 2.7 | 2447 |
| PY-$d_{10}$−BPCzA | | 3.3 | 2268 |

[a]$\Phi_P$ was calculated by the phosphorescence component in the photoluminescence spectra. If phosphorescence was too weak to be distinguished from fluorescence in the photoluminescence spectra, $\Phi_P$ was determined based on phosphorescence component associated with relevant lifetimes.
[b]No phosphorescence was detected at room temperature.
[c]Measured at 77 K.

for PY-$d_{10}$ as the guest, the $\Phi_P$ in dynamic coupling systems showed improvements of 270 to 980 times compared to the PMMA system, and 135 to 490 times when compared to β-estradiol. This substantial enhancement in quantum yield, along with the record-breaking lifetime ($\tau_{P,max} = 2447$ ms), highlighted the remarkable strength and superior effectiveness of dynamic coupling-induced RTP strategy over traditional design strategies.

## Applications

Inspired by the versatile RTP performance in the present system, multiple potential applications were explored, including photopatterning, encryption and display (Supplementary Section VIII). A 4D encryption was illustrated in Fig. 6a−c. 8 samples with various RTP colors and lifetimes were arranged to a 5 × 5 pattern, where each position was mapped to a letter at the same coordinate in the password book (Supplementary Figs. 102–107). The letters corresponding to the positions exhibiting RTP with same color were extracted to form a word. Since the pattern displayed different combinations of red, yellow and green afterglow over time, the real message was hidden in a series of fake information. The correct information 'moon' and 'lantern' could only be read out within the time range from 120 ms to 2 s.

Moreover, taking advantage of the existence of multiple phosphorescent emitters (host, guest and complex) in dynamic coupling-induced RTP systems, temperature-dependent full-color phosphorescent display was realized in a single sample of PY−BPCzA (Fig. 6d-f). As temperature increased from 60 K to 300 K, the delayed spectra showed a gradual transition from $P_H$ to $P_C$ to $P_G$. The emission color changed from blue, cyan, green, warm white, yellow, orange and red, showcasing its unique charm in thermochromic displays (Supplementary Figs. 108 and 109).

## Discussion

In conclusion, we investigated organic host−guest RTP system from a non-conventional, dynamic perspective that incorporated a delocalized complex. A dynamic coupling model was proposed, where the coupling of the complex enhanced ISC efficiency, decoupling enabled exciton transfer to $T_G$. Through this complex bridge, RTP performance surpassing the intrinsic phosphorescence of guest molecule was realized. It was noteworthy that the different behaviors between dynamic and static coupling were first clarified in the field of RTP. The necessity of dynamic coupling for facilitating RTP was emphasized, and its universality was investigated. This dynamic coupling process could be further tuned to optimize the RTP

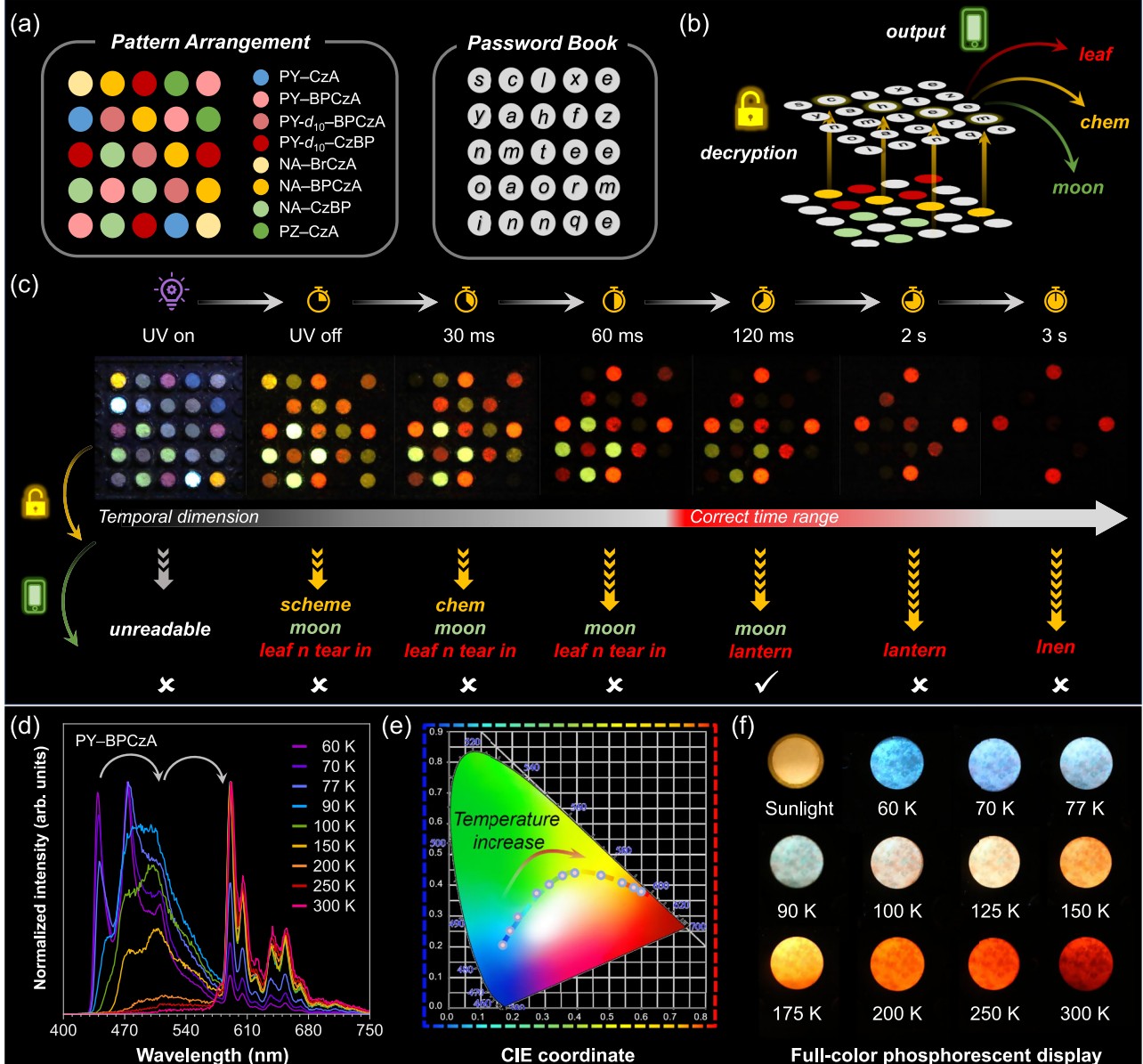

**Fig. 6 | Demonstration of potential applications.** Applications for 4D encryption (**a**–**c**) and full-color display (**d**–**f**). **a** Pattern arrangement and corresponding password book for 4D encryption. The samples used in the 5 × 5 pattern were represented by different colors. **b** Schematic diagram of the decryption process. The pattern would display different combinations of red, yellow, and green afterglow over time. The letters corresponding to the positions exhibiting afterglow with the same color were extracted to form a word. For example, here, the words formed by letters corresponding to positions exhibiting yellow, green and red afterglow were 'chem', 'moon' and 'leaf', respectively. **c** Demonstration of 4D encryption with our systems. Under UV light, the pattern emitted luminescence in various colors, so the information was unreadable. After the UV lamp was turned off, the pattern displayed readable afterglow combinations. The read-out information was written below the photos. Only the message read within the correct time range was real information. **d** Temperature-dependent delayed PL spectra of PY − BPCzA at a delay time of 10 ms, excited at 365 nm. **e** CIE coordinate diagram of delayed emission of PY − BPCzA at different temperature. **f** Photos of PY–BPCzA crystals captured after ceasing 365 nm excitation source at different temperature. The photo of PY–BPCzA crystals under sunlight was also shown.

performance. Remarkably, multicolor emissions and record-breaking red RTP with the longest lifetime of 2.4 s and $\Phi_P$ of up to 9.8% were achieved. Furthermore, we demonstrated their potential optoelectronic applications. This work clarified the nature, role, dynamics, and regulation of intermolecular coupling in the excited-state process, shedding light on the design of next-generation organic host−guest RTP systems.

## Methods
### Materials and characterization
Host molecules were synthesized in lab as shown in Supplementary Fig. 1. 1-Bromo-2-nitrobenzene (> 99.0%), benzoyl chloride (> 98.0%),

4-fluorobenzophenone (>99.0%), and tetraphenylethylene (>98.0%) were purchased from TCI (Shanghai) Development Co., Ltd. Phenylboronic acid (99.5%) was purchased from Shanghai Macklin Biochemical Technology Co., Ltd. 1-Bromohexane (≥99%), phenothiazine (≥98%), 1,8-naphthalic anhydride (≥98%), and 2,3-naphthalic anhydride (≥95%) were purchased from Aladdin Scientific. Benzo-2,1,3-thiadiazole (98%), 4,7-dibromo-2,1,3-benzothiadiazole (95%), pyrene (≥99%), and anthracene (99%) were purchased from Merck KGaA (Sigma-Aldrich). All starting materials and reagents for synthesis of host molecules were used without further purification. All guest molecules were purified through column chromatography and recrystallization twice before use.

$^1$H and $^{13}$C nuclear magnetic resonance (NMR) spectra were measured on a Bruker AVANCE III spectrometer in CDCl$_3$, with tetramethylsilane (TMS; $\delta = 0$) as the internal standard. High-resolution mass spectra (HRMS) were obtained on a GCT Premier CAB 048 mass spectrometer. High-performance liquid chromatography (HPLC) spectra were measured on an Agilent Technologies 1260 Infinity with a column of Poroshell 120 (the eluting solvent was acetonitrile/water, v/v = 9:1, flow rate = 20.0 mL/min). The single-crystal X-ray diffraction (SXRD) data were collected on a Rigaku Oxford Diffraction SuperNova with an Atlas diffractometer. During data collection, the crystals were kept at 100.01(10) K, and crystal structures were solved with Olex2. Powder X-ray diffraction (PXRD) data were collected using a PAnalytical X-ray Diffractometer (X'pert Pro). The thermogravimetric analysis (TGA) and differential scanning calorimetry (DSC) measurements were conducted on a Perkin-Elmer TGA 7 analyzer and a TA Instruments DSC Q1000, respectively, at a heating rate of 10 °C/min under nitrogen.

### General procedure for fabrication of host−guest RTP systems

99.0 mg host matrix and 1.0 mg guest were dissolved in 500 uL DCM solution. The solution was slowly dropped on a slide of glass substrate under heating at 40 °C to remove the solvent and well-dispersed doping powders were obtained. The mixed powders were then heated to the melting points of the host molecules. After melting completely, the samples were slowly cooled down to room temperature to form white solid solution of host−guest systems.

### Photophysical property measurement

Photoluminescence spectra, phosphorescence spectra, excitation-phosphorescence mapping were measured using an Edinburgh FLS 980 fluorescence spectrophotometer with a xenon arc lamp (Xe900) and a microsecond flash-lamp (μF900) as the excitation source, or a Horiba Scientific Fluorolog-3 spectrofluorometer. The lifetime curves, time-resolved emission spectra (TRES) were measured on an Edinburgh FLS980 photoluminescence spectrometer with a microsecond flash-lamp (μF900) or a picosecond pulsed diode laser as the excitation source. Absolute photoluminescence quantum yields were measured using an absolute PL quantum yield spectrometer (Hamamatsu, C11347 Quantaurus_QY). UV-Vis spectra were measured on a SHIMADZU UV-2600i UV-Vis spectrophotometer.

### Transient absorption (TA) spectroscopy

The femtosecond transient absorption (fs-TA) spectra were measured using the Helios pump-probe transient absorption spectrometer system (Ultrafast Systems, USA) equipped with the Spitfire Pro regenerative amplified Ti:sapphire laser system (Spectra Physics, USA). The 800 nm laser light, with a 120 fs pulse width, was split into two beams for pumping and probing. The pump beam was converted to a 267 nm pump beam through harmonic generation (the third harmonic of the 800 nm fundamental). The probe beam passed through a sapphire crystal, generating a white-light continuum spanning from 450 nm to 750 nm. The time-delayed probe beam, controlled by the optical delay rail with a maximum temporal delay of 3000 ps, passed through the samples and the signals were then collected by the detector. A reference probe beam was also employed to optimize the signal-to-noise ratio. The solution samples were prepared with an absorbance of approximately 1.0 at 267 nm and measured in a 2 mm path-length quartz cuvette. The solid samples were prepared by melting 1 mg sample between two quartz slides (2 cm × 2 cm) and rapidly cooling to room temperature.

The spectrometric data were recorded as a 3D wavelength-time-absorbance matrix. Prior to analysis, background subtraction, scattering light subtraction, and chirp correction were applied to all the data. The single-wavelength kinetic fitting was carried out by Surface Xplorer 4.5.

### Pressure-dependent photoluminescence measurement

High-pressure experiments were conducted using a symmetric diamond anvil cell (DAC) with culets size of 400 μm. The sample was loaded into a chamber with a diameter of approximately 130 μm in a T301 stainless steel gasket, which was preindented to a thickness of 40 μm. A small ruby ball was placed in the sample compartment for in situ pressure calibration, and the pressure was determined using the standard ruby fluorescent technique. Silicon oil served as the pressure transmitting medium (PTM) for in situ high-pressure photoluminescence spectroscopy. Such PTM did not have any detectable effect on the behavior of host−guest samples under pressure. The 355 nm line of violet diode laser was used as the excitation source. Optical photographs were recorded by a Canon EOS 5D Mark II camera. All measurements were performed at room temperature.

### Demonstration of 4D encryption and full-color display

Samples used in the applications were first prepared with a similar melt-casting process. For 4D encryption, after grinding, samples were filled into an acrylonitrile butadiene styrene (ABS) plastic plate with 5 × 5 circular holes (0.5 cm in diameter), according to the pattern arrangement shown in Fig. 6a.

For temperature-dependent full-color display, the samples were fixed between one copper plate connecting to a temperature controller and cooling system, and one quartz lid permitting the transmission of light. The holder was then placed in a vacuum chamber to avoid the condensation and ice formation of water vapor at low temperature. Optical photographs were captured by a Cannon EOS 7D camera positioned outside the quartz window of the chamber.

## Data availability

The crystallographic data for BrCzA, CzBP and BPCzA have been deposited at the Cambridge Crystallographic Data Centre (CCDC) under deposition numbers CCDC 2368574 (BrCzA), CCDC 2368575 (CzBP), and CCDC 2368576 (BPCzA), respectively. The single-crystal structure of CzA was obtained from a previous deposition, CCDC 1833958 (CzA). These data files can be obtained free of charge from CCDC via http://www.ccdc.cam.ac.uk/data_request/cif. Source data have been deposited in Figshare and can be obtained via https://doi.org/10.6084/m9.figshare.28917491. All data are available from the corresponding author upon request.

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

## Acknowledgements

This work was supported by the Research Grants Council of Hong Kong (16303221 to R.T.K.K. and C6014-20W to Z.G.), the Innovation and Technology Commission (ITC-CNERC14SC01 to J.S.), Shenzhen Key Laboratory of Functional Aggregate Materials (ZDSYS20211021111400001 to B.Z.T.) and the Science Technology Innovation Commission of Shenzhen Municipality (KQTD20210811090142053 and JCYJ20220818103007014 to B.Z.T.). We acknowledge Mr. Yuxiang Lyu for his assistance on CV measurement. We acknowledge Prof. Liangwei Ma and Prof. Xiang Ma for the helpful discussion.

## Author contributions

X.L.,W.L. and B.Z.T. conceived and designed the experiments. X.L. proposed the dynamic coupling model. Z.D. and D.L.P. assisted in the transient absorption measurements. J.W., G.X., and B.Z. assisted in the pressure-dependent experiments. S.H., X.O., R.T.K.K., J.S. and all other authors contributed to data analysis. X.L., W.L., J.W.Y.L. and B.Z.T. wrote the manuscript. J.W.Y.L., Z.G. and B.Z.T. supervised this work. X.L. and W.L. contributed equally to this work.

## Competing interests

The authors declare no competing interests.
