## [Transparent Peer Review file · Nature Communications]

Building bridges through dynamic coupling for organic phosphorescence

Corresponding Author: Professor Ben Zhong Tang

Version 0:

Reviewer comments:

Reviewer #1

(Remarks to the Author)

Recently, research devoted much effort to investigate room temperature phosphorescence (RTP) in organic materials. However, the RTP mechanism is under debated, especial in host-guest system. In this manuscript, Li and coauthors proposed a dynamic coupling model for understanding RTP. The mechanism seems reasonable, but it doesn't stand up to scrutiny. Compared with my previous work, there is not much improvement. Therefore, I do not recommend it for publication now.

1. The XRD before and after doping is not as described in manuscript, of which both are very different in Supplementary Figure 33. Please check carefully.
2. What I question most is the theoretical simulation data. The theoretical computational model described in this paper is not credible because the doping system cannot judge the stacking between the host and guest molecules. Therefore, Therefore, the relevant calculations are not reliable.
3. I have some confusion about complex (Lc) in PY-CzA system. From the study on temperature dependent spectroscopy (Supplementary Figure 40), in my opinion, I speculate that the emission ranging from 440 to at 540 nm is a mixture of the phosphorescence of the host molecules and the fluorescence of the guests in molecular or aggregate states.
4. The experiment of photoluminescence by pressure is interesting. However, it is difficult to determine the evolutionary process of dynamic coupling between the host and guest molecules, not to mention how difficult it is to quantify the stacking between the host and guest molecules.
5. The structure-activity relationship between host/guest molecular structures and dynamic coupling ability is not found, which is very helpful to understand the mechanism.

Reviewer #2

(Remarks to the Author)

The authors report on a complex-based approach to enhance room temperature phosphorescence (RTP) of organic compounds in solid samples deposited by a melt-casting method. The observed complex is like an exciplex (an intermolecular charge transfer state). Nevertheless, the authors state that there are differences between exciplexes and the observed complex. The authors conducted numerous experimental and theoretical studies to investigate the mechanism of the complex-based approach for enhancing RTP. Generalising the proposed approach, the conditions (numerous host: guest systems) under which RTP properties are most effective and when the exciplex-based approach is not effective are identified. Finally, the dynamic coupling model of RTP enhancement is proposed. The proposed complex-based approach is similar to the exciplex-based approach proposed for long-pristine luminescence [Nature, 2017, 550(19), 584–587. doi:10.1038/nature24010; Adv. Mater. 2018, 30, 1800365. Doi: 10.1002/adma.201800365; Adv. Mater. 2018, 1803713. Doi: 10.1002/adma.201803713; Nat. Commun. 2020, 11, 191 Doi:10.1038/s41467-019-14035-y; Adv. Funct. Mater. 2020, 2000795. Doi: 10.1002/adfm.202000795]. Quantum yields of RTP of the studied host-guest systems do not exceed 15 % (Supplementary Table 4). Some statements still require revisions. Despite the mentioned weaknesses, the manuscript can be appropriate for publication in Nat. Commun after the appropriate amendment.

Comments.

1. It should be demonstrated whether the complex-based approach is more efficient compared to the polymeric host-based and/or crystallization-based approaches for RTP enhancement. To do that, the RTP quantum yields of the most efficient RTP emitter (e.g. PY-d10) should be recorded in polymeric host (e.g. PMMA) and/or amorphous host (e.g. PY-d10(1 wt.

%)–BrCzA). I suggest to shift a part of Supplementary Table 4 to the main text, including into this table the most important photophysical data.

2. Supplementary Figure 42 shows the UV-Vis absorption spectrum of the system PY–CzA. This spectrum is characterized by a low-energy band attributed to the complex (but not exciplex formed between PY and CzA) observed in the ground state. To be sure that this band is not related to the RTP emitter PY, the absorption spectrum of the neat film of the emitter PY should be added to Supplementary Figure 42.

3. The excitation spectra of the neat films of the host CzA and guest PY should be added to Figure 4a.

4. Taking into account that intermolecular charge transfer (exciplex-like) can be observed for ground state [“A simple method to measure intermolecular charge-transfer absorption of organic films.” *Org. Electron.* 2018, 62, 511–515], the differences between exciplexes and the observed complex should be discussed in more detail.

5. If static and dynamic complexes are observed (Figure 4a), it is not clear why only dynamic coupling is mentioned in the title. Is the effect of the formation of the static complex negligible in RTP enhancement?

6. The previously proposed “dynamic” state-energy diagrams should be mentioned in the introduction [10.1016/j.cej.2020.127902]. In addition, examples of the recent most efficient RTP emitters should be mentioned [10.1039/D3TC04514E or 10.1021/acssuschemeng.3c04011].

Reviewer #3

(Remarks to the Author)

In this work, Li et al. discovered a dynamic coupling interaction in host-guest doping systems that enhanced room temperature phosphorescence, which was a highly innovative finding. They captured the previously overlooked excited-state coupled intermediate and provided a detailed mechanism investigation on the underlying photophysical process. More importantly, they proposed the different behaviors between dynamic and static coupling in the excited state through theoretical simulations and pressure-dependent experiments, demonstrating the necessity of dynamic process. This mechanism model exhibited broad universality across different systems, and they successfully regulated the dynamic coupling process to further enhance RTP performance. Overall, this work resolved the long-standing issue of unclear mechanisms in host-guest RTP systems, proposed a new mechanism model, and achieved an exceptional record for red RTP. Therefore, I believe that it will attract significant attention in the fields of photophysics and material science, and is worthy of publication in Nature Communications after minor revision.

1. What is the design principle for dynamic coupling based RTP system? How to choose suitable host and guest molecules?
2. What is the quantum yield of the host molecules used in the present system, and how does it change after doping guest molecules?
3. The phosphorescence spectra of PZ, NA, BDZ, AN used in guest engineering should be supplemented to confirm the origin of the RTP.
4. For host engineering, why is the lifetime of PY-BPCzA shorter than PY-CzBP?
5. In Figure 6f, achieving full-color RTP in one sample is very interesting, but the term ‘sunlight’ should be added below the first photo for clarification.

Version 1:

Reviewer comments:

Reviewer #1

(Remarks to the Author)

In the revised manuscript, the authors have implemented certain modifications and provided clarifications in response to the feedback, resulting in a noticeable enhancement in the overall quality of the work. Nevertheless, I continue to find the responses to comments 2 and 4 unsatisfactory, particularly with respect to the construction of the host-guest interaction model, which I consider to be fundamentally flawed. In the absence of a precise stacking model, the computational data presented lacks credibility. I strongly advise the authors to remove the associated data from the manuscript. Additionally, regarding the dynamic coupling interactions between the host and guest molecules, I recommend that the authors explore the development of empirical or semi-empirical physical parameters to quantitatively characterize the strength of these interactions, as opposed to relying on a macroscopic and qualitative assessment. Therefore, I still maintain my stance that the manuscript, in its present form, is not suitable for publication.

Reviewer #2

(Remarks to the Author)

Our comments are logically addressed. We suggest to accept the manuscript.

Reviewer #3

(Remarks to the Author)

The authors have well addressed my concerns. I think the quality of the manuscript is significantly improved. Thus, I recommend the acceptance of the manuscript by the journal as is.

Version 2:

Reviewer comments:

Reviewer #3

(Remarks to the Author)

[Note from the Editor: Reviewer #3 assessed the response given to reviewer #1].

I think the authors have well addressed the review's concerns, and the quality of the manuscript is significantly improved. Thus, I recommend the acceptance of the manuscript by the journal as is.

Point-by-point response to the reviewers' comments:

(Reviewers' comments and suggestions: in black; Responses to the comments and suggestions: in blue)

Reviewer: 1

Recently, research devoted much effort to investigate room temperature phosphorescence (RTP) in organic materials. However, the RTP mechanism is under debated, especial in host-guest system. In this manuscript, Li and coauthors proposed a dynamic coupling model for understanding RTP. The mechanism seems reasonable, but it doesn't stand up to scrutiny. Compared with my previous work, there is not much improvement. Therefore, I do not recommend it for publication now.

Response: We would like to thank you for your critical comments and constructive feedback on our work, which enabled us to prepare a greatly improved manuscript. The photophysical mechanism in host–guest RTP systems are indeed a topic of ongoing debate, underscoring the significance of our research. We appreciate your assessment of our mechanism as reasonable; however, we respectfully disagree with your assertion that this work lacks scrutiny and that it does not demonstrate improvement compared to previous studies.

Firstly, the dynamic coupling-induced phosphorescence was thoroughly and rigorously proved through both experimental analysis and theoretical calculations and followed a clear logic regarding existence, function, necessity, regulation and application. The **existence** of the coupling complex was confirmed through direct observation of spectral evidence. Femto-second transient absorption spectroscopy, time-resolved photoluminescence spectroscopy, lifetime monitoring, and temperature-dependent experiments were employed to investigate the photophysical process, illustrating the **function** of the coupling complex as a bridge to facilitate RTP. Importantly, the **necessity** of dynamic coupling was validated in detail through excitation spectra analysis, excitation-phosphorescence mapping, and pressure-dependent experiments. Furthermore, the **universal applicability** of dynamic coupling-induced phosphorescence was verified across 17 different host–guest systems, and the corresponding structure-property relationship was summarized. Based on this, the **regulation** of dynamic coupling process was achieved through strategic modification of host molecules, which further enhanced the RTP performance.

In addition to the experimental evidence, theoretical calculation was also carried out to support our mechanism, such as elucidating the nature of the coupling (intermolecular charge transfer), demonstrating its function (facilitating ISC), and visualizing the dynamic coupling process. Thus, from both experimental and computational perspectives, we believe this work is robust and comprehensive.

Secondly, we would like to emphasize our improvements compared to previous research. Previously, some studies proposed an intermediate state that would facilitate RTP in host–guest RTP systems (*Nat. Commun.* **2019**, *10*, 5161; *Angew. Chem. Int. Ed.* **2022**, *61*, e202200546; *J. Phys. Chem. Lett.* **2023**, *14*, 6927–6934). However, these studies either remained at the level of theoretical calculations or indirectly suggested the acceleration of ISC through comparisons of ultrafast spectra, lacking

direct observation of the complex emission in the spectroscopy. Additionally, there have been few works that successfully captured the complex emission in the PL spectra (*J. Phys. Chem. Lett.* **2021**, *12*, 4600–4608; *Angew. Chem. Int. Ed.* **2023**, *62*, e202312627). Unfortunately, these studies did not provide sufficient experimental evidence for the energy transfer process involving the complex as a bridge (transient absorption spectra, variable temperature experiments, etc.), nor did they consider the distinction between dynamic coupling and static coupling, which led to contradictions in their conclusions. Moreover, utilizing the dynamic coupling-induced phosphorescence, the RTP performance in our systems surpassed previous work, achieving multicolor emission even in the near-infrared region (800 nm), and the **longest** red RTP reported so far ($\tau_{p,\text{max}} = 2.4 \text{ s}$).

Last but not least, we apologize for any disputed descriptions in our manuscript that may have led to some of the evidence being unclear. We took your feedback into consideration and made significant improvements to our manuscript. Your comments on the experimental data (Q1 and Q3) motivated us to remeasure certain samples and supplement essential data, providing clearer evidence that the molecular packing was not affected by doping and offering stronger support for the origin of L_C. We also incorporated your feedback on theoretical calculation (Q2) to clarify the visualization of dynamic coupling processes. Furthermore, your comments on pressure-dependent experiments (Q4) encouraged us to conduct pressure-dependent XRD measurements under varying external pressure, which enhanced our understanding of the structural evolution of dynamic coupling complex. Additionally, as you suggested (Q5), the structure-activity relationship was summarized and supplemented in both the main text and the Supplementary Information.

We sincerely express our gratitude to your thorough review and would like to highlight the novelty and advancement of our work for your reconsideration. We would appreciate it if you could reconsider our manuscript and are happy to address any other concerns you may have.

(1) **Puzzle resolved:** The ‘ghost’ intermediate in previous work was directly observed as host–guest complex through spectroscopic evidence and the underlying photophysical process for facilitating RTP was clarified through robust experimental evidence.

(2) **New mechanism:** A dynamic coupling model was proposed to highlight the influence of excited-state dynamic behaviors on RTP. For the first time, the difference between dynamic and static

coupling was clarified in the field of RTP.

(3) **Advanced performance:** Dynamic coupling-induced RTP exhibited excellent universal applicability, achieving multicolor emissions and outstanding performance, including record-breaking red RTP ($\tau_p \leq 2.4$ s, $\Phi_p \leq 9.8\%$).

1. The XRD before and after doping is not as described in manuscript, of which both are very different in Supplementary Figure 33. Please check carefully.

Response: We would like to thank you for your valuable comments on the comparison of PXRD patterns before and after doping. We carefully checked the spectra in Supplementary Fig. 33 and presumed that your concern might arise from Supplementary Fig. 33b and 33c, where the BrCzA system and CzBP system seemed to be different after doping. However, their molecular packing actually remained unchanged after doping. The seemingly different pattern was due to insufficient grinding of the host molecule crystals, which led to a loss of random orientations of the crystallites. This resulted in certain diffraction peaks being too intense, hiding other peaks. Thus, we amplified the spectra of the host molecules (increasing the intensity by 6 times), allowing previously hidden peaks to become visible. As shown in Figure R1, the diffraction patterns of the doping systems PY-BrCzA and PY-CzBP aligned very well with those of the host molecules, indicating that they had the same crystal packing.

Figure R1 | PXRD patterns of BrCzA crystals (a), CzBP crystals (b), before and after doping with PY molecules. To better examine the diffraction patterns of host molecules, their spectra with the intensity multiplied by 6 were included for reference. The peaks at 8.8° , 18.0° , 26.2° , 26.8° in BrCzA $\times 6$, as well as peak at 23.3° in CzBP $\times 6$, were truncated due to exceeding the intensity range. Green dotted reference lines have been added for clearer comparison.

To provide a clearer and more intuitive comparison of the PXRD patterns, we ground the crystals more thoroughly and remeasured some samples in Supplementary Fig. 33. As a result, the diffraction patterns in the doping systems were clearly consistent with those of the host crystals, demonstrating that doping with a guest ratio of 1.0% did not change the molecular packing of the host crystals.

Supplementary Figure 33 | PXRD patterns of host-guest crystals with PY, NA and PZ as guests, respectively. CZA system (a), BrCZA system (b), CzBP system (c), BPCZA system (d). PXRD patterns of corresponding host molecules were displayed for reference.

Thus, we updated the PXRD patterns, and our changes made to the revised Supplementary Information are highlighted in yellow as follows:

On page 21 in the revised Supplementary Information:

Supplementary Figure 30 | PXRD patterns of host molecules, CZA (a), BrCZA (b), CzBP (c), BPCZA (d). Simulated patterns based on their single crystals were attached for reference.

On page 24 in the revised Supplementary Information:

Supplementary Figure 33 | PXR D patterns of host–guest crystals with PY, NA and PZ as guests, respectively. CzA system (a), BrCzA system (b), CzBP system (c), BPCzA system (d). PXR D patterns of corresponding host molecules were displayed for reference.

2. *What I question most is the theoretical simulation data. The theoretical computational model described in this paper is not credible because the doping system cannot judge the stacking between the host and guest molecules. Therefore, Therefore, the relevant calculations are not reliable.*

Response: Thank you for your thorough review and critical comments. We made extensive efforts to determine the accurate conformation of host–guest complex in the doping systems but none of them worked. Firstly, cocrystals of host and guest molecules were cultivated with different ratio, solvent, temperature and methods. Unfortunately, no cocrystals was obtained. Even if cocrystals could be obtained, it would be inaccurate to assume that the conformation in the doping system was the same as that of the cocrystals. Furthermore, single-crystal X-ray diffraction analysis was carried out for both pure host crystals and doped crystals, but their crystal structures remained the same, due to the low doping concentration. This was consistent with the unchanged PXRD patterns. Thus, it was difficult to characterize the stacking between the host and guest molecules through experimental analysis.

However, this issue was not unique to our work. It was a common challenge encountered in all doping systems. To our knowledge, no one had successfully identified the stacking between host and guest molecules in doping systems through experimental methods. Many insightful studies had simulated the stacking by optimizing the conformation of dimers using theoretical calculations, extending beyond the field of RTP. For instance, Prof. Guoqing Zhang et al determined the molecular stacking between host–guest dimer in doping RTP system through theoretical calculation, to study the importance of disorder in host molecules for facilitating RTP (*Angew. Chem. Int. Ed.* **2023**, *62*, e202312627). They also calculated the intermolecular interactions based on optimized host–guest dimer conformation in photo-induced CT complex systems (*Chem* **2024**, *10*, 2829–2843). Similarly, Prof. Bin Liu et al simulated the stacking between host and guest molecules in doping RTP and mechanoluminescence systems, studying their corresponding electron-hole distributions (*Angew. Chem. Int. Ed.* **2023**, *62*, e202310335; *Nat. Commun.* **2024**, *15*, 3668). Prof. Xiaogang Liu et al optimized the conformation of dimers in the doping RTP systems and investigated their NTO for different excited states (*Angew. Chem. Int. Ed.* **2022**, *61*, e202200546). Our group also utilized theoretical calculations to simulate dimer conformations and calculate the corresponding SOC matrices (*Matter* **2022**, *5*, 3499–3512; *Adv. Mater.* **2021**, *33*, 2007811). Additionally, Prof. Li Dang's recent work employed a similar calculation strategy to optimize the conformation of host–guest dimers (*Nat. Commun.* **2024**, *15*, 4674). Thus, optimizing the conformation of host–guest dimers through theoretical calculations could be served as a credible method to simulate the stacking between host and guest molecules in doping systems.

In our work, although we could not confirm the accurate conformation of the host–guest complex, the observation that the transition from S_C/T_C to the T_G state involved a dissociation trend was a universal finding, which had been validated in PY–CzA, PY–CzBP, and PY–BPCzA (see Figure 4c and Supplementary Figs. 43, 76, 77 for further details). Additionally, the significant misalignment observed in the complex during the transition from S_C state to T_G state in Prof. Guoqing Zhang's study also supported this viewpoint (*Angew. Chem. Int. Ed.* **2023**, *62*, e202312627). Thus, rather than determining the precise stacking between host and guest molecules, our theoretical calculations focused on demonstrating that the host–guest complex needed to undergo a conformational change

in the excited states, particularly a dissociation process, to reach the T_G state. This process was what we referred to as the dynamic coupling process.

Furthermore, our theoretical calculations served solely as a reference tool to visualize the dynamic processes of the complex in the excited states. The mechanism of dynamic coupling-induced phosphorescence was, in fact, supported by robust experimental evidence:

- (1) Ultrafast transient absorption spectroscopy, time-resolved photoluminescence spectroscopy, and lifetime monitoring, along with variable temperature experiments, confirmed the photophysical pathway of excitons transitioning from S_H to S_C , followed by ISC to T_C , and finally to T_G (Figure 3f).
- (2) Excitation spectra analysis and excitation-phosphorescence mapping clearly showed that phosphorescence originated from dynamic coupling, while static coupling tended to trap excitons (Figure 4e).
- (3) Pressure-dependent experiments showed increasing pressure hindered the dynamic coupling process and consequently suppressed phosphorescence, providing further evidence for the necessity of dynamic coupling in facilitating phosphorescence.

To sum up, we explored various methods to obtain the accurate conformation of the complex through experimental analysis but were unsuccessful. Meanwhile, previous research indicated that theoretical calculations could be a reliable approach to investigate the stacking between host and guest molecules. Additionally, the simulated dynamic process was a universal observation in all three systems in our work and was also found in other reported systems. Furthermore, we used theoretical calculations primarily as a reference tool to visualize the dynamic processes, rather than relying on them to determine the precise conformation. More importantly, the mechanism of dynamic coupling-induced phosphorescence was validated by robust experimental evidence, not solely by theoretical simulations. We appreciate your comments about theoretical calculation, and we would be more than willing to try if you have any suggestions on obtaining the stacking between the host and guest molecules through experimental techniques.

More details regarding the decoupling process in PY/CzA, PY/CzBP, and PY/BPCzA, as visualized by theoretical calculations, were provided below.

In PY/CzA, compared to S_0 , complex at S_C and T_C showed increased proximity and overlap of CzA and PY, indicating the occurrence of coupling process. However, in the T_G state, CzA and PY were noticeably misaligned, which significantly reduced the extent of overlap and increased the center-to-center distance (Figure 4c). This demonstrated a decoupling process. The intermolecular noncovalent interactions (NCI) analysis also revealed the presence of stronger π - π interactions at S_C and T_C compared to S_0 , which became weakened after the transition to T_G (Supplementary Fig. 43).

Figure 4c | Optimized conformations of PY/CzA dimer in the ground state (S_0), singlet state of complex (S_C), triplet state of complex (T_C), and triplet state of guest (T_G), from the top view (top) and side view (bottom). Color code: blue, CzA; red, PY. Hydrogen atoms and hexyl groups were hidden for clear visualization. Overlap areas between aromatic ring planes were colored in light green. Overlap ratios and center-to-center distances were labeled.

Supplementary Figure 43 | The distributions of NCI regions in PY/CzA dimer in the different states from the top view (left) and side view (right). Color code: ice blue, C; gray, H; red, O. The green areas represented the π - π interactions between the phenyl ring and carbazole moiety.

In PY/CzBP, the overlap ratio increased, and the π - π interaction region expanded from S_0 to S_C state, indicating the coupling process occurred. While from T_C to T_G , the overlap ratio decreased and the NCI region shrank, corresponding to a dissociation trend.

Supplementary Figure 76 | The optimized conformations of PY/CzBP dimer (left) and the distributions of NCI regions (right) in the different states. Color code in the conformations: blue, CzBP; red, PY. Hydrogen atoms were hidden for clear visualization. The overlap areas between aromatic ring planes were colored in light green. The overlap ratios were calculated by the proportion of the overlap areas to the areas of the carbazole moiety and labeled on the left side. Color code in the NCI analysis: ice blue, C; gray, H; red, O. The green areas represented the π - π interactions between the phenyl ring and carbazole moiety.

In PY/BPCzA, the process was similar to PY/CzA and PY/CzBP, except the overlap ratio in S_C decreased. This was due to the strong C-H \cdots O interaction between the pyrene ring and carbonyl group on BPCzA (the blue area in Supplementary Fig. 77). The enhanced C-H \cdots O interaction suggested the coupling process also occurred in the S_C state. And the decreased overlap ratio and shrunk NCI region from T_C to T_G state were consistent with the dissociation process.

Supplementary Figure 77 | The optimized conformations of PY/BPCzA dimer (left) and the distributions of NCI regions (right) in the different states. Color code in the conformations: blue, BPCzA; red, PY. Hydrogen atoms and hexyl groups were hidden for clear visualization. The overlap areas between aromatic ring planes were colored in light green. The overlap ratios were calculated by the proportion of the overlap areas to the areas of the carbazole moiety and labeled on the left side. Color code in the NCI analysis: ice blue, C; gray, H; red, O. The green areas represented the π - π interactions between the phenyl ring and carbazole moiety. The blue areas represented the C-H \cdots O interactions.

3. I have some confusion about complex (Lc) in PY-CzA system. From the study on temperature dependent spectroscopy (Supplementary Figure 40), in my opinion, I speculate that the emission ranging from 440 to at 540 nm is a mixture of the phosphorescence of the host molecules and the fluorescence of the guests in molecular or aggregate states.

Response: Thanks for your careful review and constructive comments. We fully agreed that the emission ranging from 440 to 540 was a mixture and one component should be phosphorescence of host molecules, and we are sorry for the unclear description. However, the second component here was not fluorescence of guest in molecular or aggregate states, instead, it was the complex emission. The reasons were listed below.

(1) The spectra in Supplementary Fig. 40 were all measured after a delay of 10 ms. The lifetime of fluorescence of pyrene either in molecular state or aggregate state was much shorter than this range (Figure R2), so it had already been cut off by spectrometer. Instead, as shown in Figure R3, the emission ranging from 440 to 540 exhibited lifetime in the millisecond range from 77 K to 300 K. As temperature increased, the lifetime gradually decreasing, ruling out the possibility of TADF and proving the emission originated from a triplet state. Therefore, the second component in this case was not from the fluorescence of pyrene in molecular state or aggregate state. Instead, it should originate from a triplet state with a lifetime in the millisecond range.

Figure R2 | Lifetime curves of PY-PMMA recorded at 390 nm (a) and PY crystals recorded at 475 nm (b). Fitted lifetime values were indicated.

Figure R3 | Lifetime curves of PY-CzA at variable temperature with emission recorded at 470 nm. As temperature increased, the lifetime showed an obviously decreasing trend due to thermal quenching.

(2) The emission spectrum of the second component was different from PY in molecular state or aggregate state, but it was consistent with the complex emission spectrum. As indicated in Supplementary Fig. 33a, the phosphorescence spectrum of the host crystal was also observed in the range of 440-540 nm. However, the discrepancy between this spectrum and the delayed spectra of PY-CzA measured at 77 K in Supplementary Fig. 40 suggested the existence of a second component. As the temperature increased, the non-radiative decay from T_1 state of host molecules became more pronounced, resulting in the disappearance of its phosphorescence when temperature reached 200 K. At this point, the second component could be differentiated and was distinctly observable (Figure R4a). It appeared as a broad peak with a maximum around 470 nm, which was consistent with the complex emission observed in the prompt PL spectra of PY-CzA (Figure R4b). However, it was clearly different from PY in both molecular and aggregate states (Figure R4c and R4d). In the molecular state, the emission was centered at 400 nm, while in the aggregate state, the emission was located within a similar range but exhibited structured peaks. Thus, the second component here was complex emission, not fluorescence of guest in molecular or aggregate states.

Figure R4 | (a) Delayed PL spectra of PY-CzA at a delay time of 10 ms measured at 77 K, 200 K and 300 K, respectively. (b) Prompt PL spectra of PY-CzA with complex emission L_C noted. (c) Prompt PL spectra of PY-PMMA film. (d) Prompt PL spectra of PY crystals.

(3) The excitation spectrum of PY-CzA recorded at L_C differed from that of PY in either its molecular or aggregate state (Figure R5). This further ruled out the possibility that the emission in the 440-540 nm range originated from PY. Instead, it resulted from a combination of dynamic and

static coupling, as discussed in Figure 4a of the main text.

Figure R5 | (a) Comparison between excitation spectrum of PY-CzA measured at L_C , excitation spectra of PY measured at molecular state and aggregate state. For molecular state, excitation spectrum of PY-PMMA was measured with emission recorded at 410 nm. For aggregate state, excitation spectrum of PY crystals was measured with emission recorded at 475 nm. (b) Analysis of excitation spectrum of PY-CzA measured at L_C . The excitation spectrum of the complex emission was measured at 470 nm (L_C ; blue solid line). The dynamic complex component was shown in blue line with circles. The static complex component was shown in green line with circles.

In summary, the measurement of lifetime, comparison of emission spectra, and analysis of excitation spectra all demonstrated that the delayed emission in Supplementary Fig. 40 did not include the fluorescence of the guest in either its molecular or aggregate state. Furthermore, the consistent emission spectra and similar excitation spectra supported the conclusion that the emission ranging from 440 to 540 nm at low temperature was a mixture of phosphorescence from the host molecules and complex emission.

Thus, we revised some sentences to give a more accurate description, and our changes made to the revised manuscript and Supplementary Information are highlighted in yellow as follows:

On page 6 in the revised manuscript:

To further prove the occurrence of ET process, temperature-dependent delayed PL spectra were measured (Fig. 3e). At low temperature, PY-CzA exhibited a mixture of phosphorescence of host crystals, P_H , and complex emission, L_C , in the 420-540 nm range, along with a weak phosphorescence of guest molecules in the 590-750 nm range. As the temperature increased to 200 K, P_H disappeared due to thermal quenching and L_C became distinctly observable. At the same time, the intensity of phosphorescence of guest, P_G , started to increase, indicating that the presence of complex favored the formation of phosphorescence. As the temperature continued to rise, instead of being weakened by enhanced thermal vibration, P_G significantly intensified while L_C diminished, confirming the existence of a thermally activated ET process from T_C to T_G (Supplementary Fig. 40). Therefore, the photophysical process involved in the coupling-induced phosphorescence was

summarized in Fig. 3f.

On page 30 in the revised Supplementary Information:

At 77 K, PY-CzA exhibited two emission bands: one located between 420 and 540 nm, which was a mixture of phosphorescence from host crystals (P_H) and complex emission (L_C); and the other between 590 and 750 nm, originating from the phosphorescence of guest molecules (P_G). The non-radiative transitions from the triplet states were sufficiently suppressed at low temperature, resulting in P_H dominating at 77 K, while P_G was weak due to the low doping concentration of PY and its poor ISC ability. As temperature increased, both P_H and P_G slightly decreased due to the enhanced thermal vibrations. When the temperature increased to 200 K,

Supplementary Figure 40 | Variable temperature study on PY-CzA system. (a) Temperature-dependent delayed PL spectra of PY-CzA at a delay time of 10 ms, excited at 365 nm. (b) The intensity of L_C (470 nm) and P_G (595 nm) at different temperature. (c) Lifetime curves of P_G at different temperature, recorded at 595 nm. (d) Schematic diagram of thermally activated ET process during the variable temperature experiment in PY-CzA system.

On page 8 in the revised manuscript:

To further investigate the excited-state behaviors of complex involved in this coupling-induced phosphorescence, excitation spectrum of PY-CzA was measured at L_C (Fig. 4a). It was different from the excitation spectra of PY-PMMA film and PY crystals, excluding the possibility that L_C originated from guest molecules (Supplementary Fig. 41). From 300 nm to 373 nm, the spectrum resembled that of CzA crystals, indicating an exciplex formed through coupling excited-state CzA

and ground-state PY.

On page 31 in the revised Supplementary Information:

Supplementary Figure 41 | Comparison between excitation spectrum of PY-CzA measured at L_C , excitation spectra of PY measured at molecular state and aggregate state. For molecular state, excitation spectrum of PY-PMMA (1.0 wt%) was measured with emission recorded at 410 nm. For aggregate state, excitation spectrum of PY crystals was measured with emission recorded at 475 nm.

4. The experiment of photoluminescence by pressure is interesting. However, it is difficult to determine the evolutionary process of dynamic coupling between the host and guest molecules, not to mention how difficult it is to quantify the stacking between the host and guest molecules.

Response: We appreciated your interest and comments on the pressure experiments. Although there was a lack of techniques to accurately determine the stacking between host and guest, high pressure experiments provided a mature method to determine the structural evolution in host–guest systems.

Firstly, previous studies had utilized high-pressure X-ray diffraction or infrared spectroscopy to quantify the evolutionary process in the structure of organic crystals, including interplanar spacing, unit cell volume, bond strength, etc. (*J. Am. Chem. Soc.* **2024**, *146*, 28961–28972; *Nat. Commun.* **2022**, *13*, 5234; *J. Am. Chem. Soc.* **2020**, *142*, 1153–1158). Inspired by these work, we conducted *in situ* pressure-dependent X-ray diffraction to monitor the interplanar spacing of host–guest systems under external pressure.

As shown in Supplementary Fig. 46a, all diffraction peaks shifted toward higher angles with increasing pressure, suggesting an obvious reduction in the unit cell volume and interplanar spacing. The diffraction peaks became wider, and the intensity kept decreasing as pressure increased, suggesting a gradual amorphization process. Taking (004) crystal plane as an example, we calculated its interplanar spacing, which showed a continuous decreasing from 4.56 Å to 4.33 Å (Supplementary Fig. 46b). Since (004) crystal plane was perpendicular to *c* axis, the compressed rate in the direction of *c* axis could be quantified accordingly. Supplementary Fig. 46c showed that the molecular packing of BrCzA exhibited a herringbone arrangement in the crystal structure of PY–BrCzA system and Supplementary Fig. 46d illustrated how the crystal was compressed in the direction of *c* axis. Thus, no matter if the doped PY molecule replaced BrCzA molecule at its original position or filled in the space between BrCzA molecules, the intermolecular distance between PY and BrCzA in the complex decreased under pressure due to a cell volume shrinkage.

The smaller distance between PY and BrCzA would increase the overlap between their molecular orbitals, leading to stronger intermolecular charge transfer (CT) interactions. This was further supported by spectral evidence showing that the complex emission L_C exhibited a gradual red-shift and weakened intensity with increasing pressure (Supplementary Figs. 47a and 47b). Surprisingly, the trend of weakening intensity of L_C was highly consistent with the trend of reduced interplanar spacing, further confirming the process of decreased intermolecular spacing and strengthened intermolecular coupling under high pressure (Supplementary Fig. 47c).

Furthermore, although less explored, previous research had indicated that in RTP systems without a coupling complex, increasing pressure typically restricted molecular motion and suppressed non-radiative decay, thereby enhancing phosphorescence (*Nat. Commun.* **2024**, *15*, 7778; *Matter* **2020**, *3*, 449–463; *Chem. Sci.* **2021**, *12*, 4425–4431). In contrast, in our system, despite the suppression of non-radiative transitions, phosphorescence gradually weakened and eventually quenched (Supplementary Fig. 47d). This was because the coupling interactions between host and guest molecules were strengthened with increased pressure. Additionally, the high pressure restricted molecular motion, hindering the dissociation of complex during dynamic coupling process. This made it difficult for excitons to reach T_G state, ultimately leading to the disappearance of

phosphorescence. Therefore, pressure-dependent experiments provided compelling evidence for dynamic coupling-induced phosphorescence through *reductio ad absurdum*.

As discussed in response to Q2, our purpose was not to obtain the accurate stacking between the host and guest but to demonstrate the necessity of dynamic coupling process. This was effectively validated by the decreased intermolecular spacing, the red-shift and weakening of the L_C, the suppression of non-radiative decay, and the atypical quenching of phosphorescence in pressure-dependent experiments.

Supplementary Figure 46 | Structural evolution of PY-BrCzA system under high pressure. (a) *In situ* pressure-dependent X-ray diffraction patterns of PY-BrCzA crystals. (b) Interplanar spacing of (004) crystal plane under different pressure. The right axis indicated the compression rate in the direction of c axis. (c) Molecular packing of BrCzA in the crystal structure. (d) Schematic diagram of compressed crystals under high pressure.

Supplementary Figure 47 | (a) *In situ* pressure-dependent PL spectra of PY-BrCzA crystals. (b) Variation of L_C in intensity and wavelength. Inset illustrated that with increasing pressure, the coupling between host and guest was strengthened. (c) Variation of L_C intensity and interplanar spacing of (004) crystal plane as external pressure increased. (d) The intensity variations of L_C and P_G as external pressure increased. Inset illustrated that as the decoupling process was suppressed, the phosphorescence was weakened and eventually quenched.

Thus, we supplemented the discussion on structural evolution of PY-BrCzA system under high pressure and provided a more detailed description of the coupling strengthened by external pressure. To enhance accuracy, we revised the inset in Figure 4d to plot the L_C peak intensity instead of indiscriminately selecting the intensity at 480 nm. Our changes made to the revised manuscript and Supplementary Information are highlighted in yellow as follows:

On page 8 in the revised manuscript:

In situ pressure-dependent X-ray diffraction showed that applying external pressure led to a tighter crystal stacking, reducing the intermolecular distance and increasing orbital overlap within complex (Supplementary Fig. 46). This would result in a strengthened coupling of complex with enhanced CT characteristic. As expected, the emission (L_C) gradually red-shifted with weaker intensity as the

pressure increased (Supplementary Figs. 47 and 48). However, contrary to the anticipated enhancement due to the suppression of molecular vibrations under high pressure⁵¹, P_G weakened and eventually disappeared (Fig. 4d).

On page 9 in the revised manuscript:

Fig. 4 | d, Pressure-dependent PL spectra of PY-BrCzA excited at 355 nm. Inset: the intensity variations of L_C and P_G as external pressure increased. The upper schematic diagram illustrated that with increasing pressure, the coupling between host and guest was strengthened and the decoupling process was suppressed. Therefore, the complex exhibited a red-shifted emission and the phosphorescence, P_G , was weakened.

On page 34 in the revised Supplementary Information:

Pressure-dependent experiments

As shown in Supplementary Fig. 46a, all diffraction peaks shifted toward higher angles with increasing pressure, suggesting an obvious reduction in the unit cell volume and interplanar spacing. The diffraction peaks became wider, and the intensity kept decreasing as pressure increased, suggesting a gradual amorphization process. Taking (004) crystal plane as an example, we calculated its interplanar spacing, which showed a continuous decreasing from 4.56 Å to 4.33 Å (Supplementary Fig. 46b). Since (004) crystal plane was perpendicular to c axis, the compressed rate in the direction of c axis could be quantified accordingly. Supplementary Fig. 46c showed that the molecular packing of BrCzA exhibited a herringbone arrangement in the crystal structure of PY-BrCzA system and Supplementary Fig. 46d illustrated how the crystal was compressed in the direction of c axis. Thus, no matter if the doped PY molecule replaced BrCzA molecule at its original position or filled in the space between BrCzA molecules, the intermolecular distance between PY and BrCzA in the complex decreased under pressure due to a cell volume shrinkage.

The smaller distance between PY and BrCzA would increase the overlap between their molecular orbitals, leading to stronger intermolecular charge transfer (CT) interactions. This was further supported by spectral evidence showing that the complex emission L_C exhibited a gradual red-shift and weakened intensity with increasing pressure (Supplementary Figs. 47a, 47b and 48). Surprisingly, the trend of weakening intensity of L_C was highly consistent with the trend of reduced interplanar spacing, further confirming the process of decreased intermolecular spacing and strengthened intermolecular coupling under high pressure (Supplementary Fig. 47c).

Furthermore, increasing pressure typically restricted molecular motion and suppressed non-radiative decay, thereby enhancing phosphorescence. However, in our system, despite the suppression of non-radiative transitions, phosphorescence gradually weakened and eventually quenched (Supplementary Fig. 47d). This was because the coupling interactions between host and guest molecules were strengthened with increased pressure. Additionally, the high pressure restricted molecular motion, hindering the dissociation of complex during dynamic coupling process. This made it difficult for excitons to reach T_G state, ultimately leading to the disappearance of phosphorescence. Therefore, pressure-dependent experiments provided compelling evidence for dynamic coupling-induced phosphorescence through *reductio ad absurdum*.

On page 35 in the revised Supplementary Information:

Supplementary Figure 46 | Structural evolution of PY-BrCzA system under high pressure. (a) *In situ* pressure-dependent X-ray diffraction patterns of PY-BrCzA crystals. (b) Interplanar spacing of (004) crystal plane under different pressure. The right axis indicated the compression rate in the direction of c axis. (c) Molecular packing of BrCzA in the crystal structure. (d) Schematic diagram of compressed crystals under high pressure.

On page 36 in the revised Supplementary Information:

Supplementary Figure 47 | (a) *In situ* pressure-dependent PL spectra of PY-BrCzA crystals. (b) Variation of L_C in intensity and wavelength. Inset illustrated that with increasing pressure, the coupling between host and guest was strengthened. (c) Variation of L_C intensity and interplanar spacing of (004) crystal plane as external pressure increased. (d) The intensity variations of L_C and P_G as external pressure increased. Inset illustrated that as the decoupling process was suppressed, the phosphorescence was weakened and eventually quenched.

5. The structure-activity relationship between host/guest molecular structures and dynamic coupling ability is not found, which is very helpful to understand the mechanism.

Response: We are sincerely grateful for your valuable comments. In Section IV of the Supplementary Information (Supplementary Figs. 49–67), we investigated 17 different host–guest systems and discussed their structure–property relationship. We apologize if this section was unclear. Herein, we reviewed different scenarios in these systems, summarized the structure–property relationship, and demonstrated how to utilize the obtained relationship to guide the design of host–guest RTP systems.

I. Analysis of 17 Host–Guest Systems

Figure R6 showed the 8 guest molecules used in the investigation. The structures of these guests exhibited varying degrees of planarity, and their triplet states had different energy levels, along with diverse electron-donating and electron-accepting tendencies. When doped into CzA or BrCzA host molecules, these systems demonstrated different performances and could be categorized into three scenarios, as illustrated below (see Figures R7–R9; PL spectra could be found in Supplementary Figs. 52–64). By analyzing these examples, we could deduce the corresponding structure–property relationships.

Figure R6 | Guest molecules used for demonstration of structure–property relationship.

In the first scenario, neither L_C nor P_G was found in these systems, indicating there was no coupling between host and guest molecules to facilitate RTP. The lack of coupling might be attributed to the non-planar structure, which hindered the proximity between the host and guest molecules (TPE), or to the inappropriate intermolecular interactions (BDZ, AN). The ISC processes in host and guest molecules themselves were inefficient. The TTET process from T_H to T_G was also inhibited due to the weak ISC ability of host and the strong non-radiative dissipation from T_H . Thus, no RTP was generated and only $F_{H/G}$ was observed (Figure R7).

Figure R7 | (a) Systems with no coupling between host and guest molecules, no RTP observed. (b) Schematic diagram for photophysical process in the first scenario.

In the second scenario, L_C appeared in the PL spectra, but P_G was not generated. This indicated coupling occurred but the ET pathway to T_G during decoupling process was inhibited. The reason was that the intermolecular interactions between host and guest molecules were too strong, forming static complexes with T_C lower than T_G . This made the ET process to T_G energetically unfavored. Thus, the excitons were trapped within the static complex and decayed from T_C (Figure R8).

Figure R8 | (a) Systems with excitons trapped within static complex, no RTP observed. (b) Schematic diagram for photophysical process in the second scenario.

In the third scenario, both L_C and P_G were observed in these systems, indicating the excitons were successfully transferred to T_G state by dynamic coupling process. This highlighted the essential role of dynamic coupling in facilitating RTP (Figure R9). In these systems, both the host and guest molecules exhibited planar structures. Furthermore, their electron-donating abilities and accepting tendencies matched well with each other, leading to attractive intermolecular interactions that allowed them to couple effectively without being overly strong. Both factors benefited the formation of dynamic coupling.

Figure R9 | (a) Systems with dynamic coupling induced phosphorescence, RTP observed. (b) Schematic diagram for photophysical process in the third scenario.

II. Structure-Property Relationship

Thus, the structure-property relationship for dynamic coupling-induced phosphorescence could be deduced as below.

(1) **Host and guest molecules should have a certain degree of planarity to enable close proximity to one another.** Since the formation of the coupling complex required the molecules to be close to a certain distance, non-planar structures like TPE hindered this.

(2) **Their electron-donating and accepting tendencies should align within a specific range to form suitably intermolecular charge transfer interactions without being too strong to trap the excitons.** As discussed in Figure 1 in the manuscript, the coupling complex was stabilized by intermolecular charge transfer process. If the interaction was too weak, there would be no coupling and no RTP (Figure R7). Conversely, if the interaction was too strong, the complex would be stabilized to a lower energy level than T_G , trapping the exciton within the complex (T_C) and resulting in no RTP (Figure R8).

III. Guiding the Design of Host–Guest RTP Systems

This structure-property relationship could guide us to regulate the RTP performance in host–guest systems. For example, in systems where excitons were trapped in complex, weakening their interactions could activate phosphorescence (Figure R10a). In PZ–BrCzA, PZ had a strong electron-donating tendency, and BrCzA acted as an electron acceptor. They formed an excessively strong complex that trapped the exciton. By weakening the electron-accepting ability of BrCA by replacing with CzA, dynamic coupling was successfully achieved, thus activating phosphorescence in PZ–CzA (Figure R10b). Conversely, in NA–CzA, NA had a strong electron-accepting tendency, and CzA worked as an electron donor. The intermolecular interaction between CzA and NA was too strong and there was no RTP in NA–CzA. By brominating CzA to weaken its electron-donating ability, BrCzA successfully facilitated dynamic coupling and activated the RTP in NA–BrCzA (Figure R10c).

Figure R10 | Guiding the design of host-guest RTP systems based on structure-property relationship. (a) Schematic diagram for regulation from static to dynamic coupling. (b) Prompt and delayed PL spectra of NA-CzA and NA-BrCzA system. (c) Prompt and delayed PL spectra of PZ-BrCzA and PZ-CzA system.

We rewrote Section IV in Supplementary Information to give a more detailed description, supplemented how we utilized the structure-property relationship to design dynamic coupling-induced RTP systems, and added the discussion on structure-property relationship to the main text to emphasize its importance. Our changes made to the revised manuscript and Supplementary Information are highlighted in yellow as follows:

On page 11 in the revised manuscript:

Design Principle for Dynamic Coupling

To enrich the emission color and expand the applicable scope.....

.....

Further analysis revealed that the different performances of these host–guest systems could be categorized into three scenarios, through which the design principle for host–guest RTP systems could be clarified (Supplementary Figs. 65 and 66). Considering that the coupling process required close proximity and matching orbital energy levels, while the decoupling process necessitated appropriate intermolecular interactions, the corresponding structure-property relationship for dynamic coupling-induced phosphorescence was deduced as follows: (1) Host and guest molecules should have a certain degree of planarity to enable close proximity to one another; (2) Their electron-donating and accepting tendencies should align within a specific range to form suitably attractive intermolecular charge transfer interactions without being too strong to trap the excitons. Based on this, we further showcased how to utilize such structure-property relationship to guide the design of host–guest RTP systems, and successfully activated RTP through regulating the coupling process in host–guest systems (Supplementary Fig. 67).

On page 53 in the revised Supplementary Information:

Summary of structure-property relationship for host–guest RTP systems

In the first scenario, neither L_C nor P_G was found in these systems, indicating there was no coupling between host and guest molecules to facilitate RTP. The lack of coupling might be attributed to the non-planar structure, which hindered the proximity between the host and guest molecules (TPE), or to the inappropriate intermolecular interactions (BDZ, AN). The ISC processes in host and guest molecules themselves were inefficient. The TTET process from T_H to T_G was also inhibited due to the weak ISC ability of host and the strong non-radiative dissipation from T_H . Thus, no RTP was generated and only $F_{H/G}$ was observed (Supplementary Figs. 65a and 66a).

In the second scenario, L_C appeared in the PL spectra, but P_G was not generated. This indicated coupling occurred but the ET pathway to T_G during decoupling process was inhibited. The reason was that the intermolecular interactions between host and guest molecules were too strong, forming static complexes with T_C lower than T_G . This made the ET process to T_G energetically unfavored. Thus, the excitons were trapped within the static complex and decayed from T_C (Supplementary Figs. 65b and 66b).

In the third scenario, both L_C and P_G were observed in these systems, indicating the excitons were successfully transferred to T_G state by dynamic coupling process. This highlighted the essential role of dynamic coupling in facilitating RTP. In these systems, both the host and guest molecules exhibited planar structures. Furthermore, their electron-donating abilities and accepting tendencies matched well with each other, leading to attractive intermolecular interactions that allowed them to couple effectively without being overly strong. Both factors benefited the formation of dynamic coupling (Supplementary Figs. 65c and 66c).

The above discussion clarified the structure-property relationship for dynamic coupling-induced phosphorescence:

(1) Host and guest molecules should have a certain degree of planarity to enable close proximity to one another.

(2) Their electron-donating and accepting tendencies should align within a specific range to form suitably intermolecular charge transfer interactions without being too strong to trap the excitons.

This structure-property relationship could guide us to design host-guest RTP systems. For example, in systems where excitons were trapped in complex, weakening their interactions could activate phosphorescence (Supplementary Fig. 67a). In PZ-BrCzA, PZ had a strong electron-donating tendency, and BrCzA acted as an electron acceptor. They formed an excessively strong complex that trapped the exciton. By weakening the electron-accepting ability of BrCA by replacing with CzA, dynamic coupling was successfully achieved, thus activating phosphorescence in PZ-CzA (Supplementary Fig. 67b). Conversely, in NA-CzA, NA had a strong electron-accepting tendency, and CzA worked as an electron donor. The intermolecular interaction between CzA and NA was too strong and there was no RTP in NA-CzA. By brominating CzA to weaken its electron-donating ability, BrCzA successfully facilitated dynamic coupling and activated the RTP in NA-BrCzA (Supplementary Fig. 67c).

Supplementary Figure 65 | Three scenarios among 17 host-guest systems.

Supplementary Figure 66 | Photophysical processes in above 17 host–guest systems.

Supplementary Figure 67 | Guiding the design of host-guest RTP systems based on structure-property relationship. (a) Schematic diagram for regulation from static to dynamic coupling. (b) Prompt and delayed PL spectra of NA-CzA and NA-BrCzA system. (c) Prompt and delayed PL spectra of PZ-BrCzA and PZ-CzA system.

Reviewer: 2

The authors report on a complex-based approach to enhance room temperature phosphorescence (RTP) of organic compounds in solid samples deposited by a melt-casting method. The observed complex is like an exciplex (an intermolecular charge transfer state). Nevertheless, the authors state that there are differences between exciplexes and the observed complex. The authors conducted numerous experimental and theoretical studies to investigate the mechanism of the complex-based approach for enhancing RTP. Generalising the proposed approach, the conditions (numerous host-guest systems) under which RTP properties are most effective and when the exciplex-based approach is not effective are identified. Finally, the dynamic coupling model of RTP enhancement is proposed. The proposed complex-based approach is similar to the exciplex-based approach proposed for long-pristine luminescence [*Nature*, 2017, 550(19), 584–587. doi:10.1038/nature24010; *Adv. Mater.* 2018, 30, 1800365. Doi: 10.1002/adma.201800365; *Adv. Mater.* 2018, 1803713. Doi: 10.1002/adma.201803713; *Nat. Commun.* 2020, 11, 191 Doi:10.1038/s41467-019-14035-y; *Adv. Funct. Mater.* 2020, 2000795. Doi: 10.1002/adfm.202000795]. Quantum yields of RTP of the studied host-guest systems do not exceed 15 % (Supplementary Table 4). Some statements still require revisions. Despite the mentioned weaknesses, the manuscript can be appropriate for publication in *Nat. Commun* after the appropriate amendment.

Response: We would like to thank you for your thorough comments and valuable suggestions, and we have carefully made a proper revision of the manuscript. These comments and suggestions have not only helped us enhance the quality of our manuscript significantly but also motivated us to pursue more comprehensive research on the dynamic coupling process in our future endeavors.

Firstly, we appreciate your interest in our mechanism and would like to clarify that our system was fundamentally different from the LPL systems you mentioned.

(1) The photophysical process was very different with different species involved (Figure R11).

The most important species in LPL systems was the charge separation (CS) state, which was radical ion pair formed after the dissociation of the charge transfer (CT) state. The CS state acted as a charge carrier, diffusing throughout the system, and served as the rate-limiting step in the entire photophysical process. Following slow charge recombination to the CT excited state, some systems decayed directly to the ground state and exhibited CT emission (*Nature*, 2017, 550(19), 584–587; *Adv. Mater.* 2018, 30, 1800365), while others transferred to the triplet state of the guest and emitted phosphorescence (*Adv. Mater.* 2018, 1803713). This depended on which state had the lower energy level (*Nat. Commun.* 2020, 11, 191; *Adv. Funct. Mater.* 2020, 2000795). In either case, both processes belonged to LPL emission, with the long lifetime originating from the diffusion of the CS state. However, there was no such charge separation process or CS state involved in dynamic coupling-induced RTP system, due to the higher combination energy of CT complex. The excitons directly transferred from dynamic coupling complex to the triplet state of guest through conformation changes. Thus, the photophysical process was totally different.

Figure R11 | Schematic diagram for photophysical process in LPL systems (a) and dynamic coupling-induced RTP systems (b).

(2) The origin of the long-lived emission (rate-limiting step) was different (Figure R12).

In LPL systems, the long-lived emission originated from the slow recombination of charge carriers, with lifetime ranging from minutes to hours (Figure R12a). However, in our systems, there was no CS state involved; instead, the rate-limiting step was the decay process from triplet state (Figure R12b). The long-lived emission originated from the long lifetime of triplet excitons, which remained in the millisecond to second range.

Figure R12 | Energy level diagram for photophysical process in LPL systems (a) and dynamic coupling-induced RTP systems (b).

(3) The decay kinetics of the long-lived emission was different (Figure R13).

Since the rate-limiting step in LPL systems was governed by the slow recombination of dissociated CS state/radical ion pairs, it aligned with diffusion physical model and its kinetics followed a power-law decay. However, our systems didn't involve CS, CR, or diffusion process; instead, the rate-

limiting step was the decay from triplet state of guest. Thus, it followed an exponential decay as other RTP systems did. Figure R13 showed a clear comparison between LPL and phosphorescence through logarithmic plot of lifetime curves (Figure R13a). It was obvious that dynamic coupling-induced RTP in our system was totally different from LPL systems (Figure R13b and R13c).

Figure R13 | (a) Different decay kinetics between LPL and phosphorescence. (b) Typical decay kinetics of LPL system. (c) Decay kinetics of dynamic coupling-induced RTP in our systems. The emission decay profiles were plotted on a log-log scale.

(4) Excitation dependence was different (Figure R14).

The long lifetime of LPL required charge accumulation, so prolonged photoexcitation would benefit the charge separation process (*Nanoscale*, 2021, 13, 8412-8417). As shown in Figure R14a, a longer LPL duration was achieved as the photoexcitation time increased. However, no such excitation time dependence was found in our system (Figure R14b). This further proved that the dynamic coupling-induced RTP was essentially different from LPL systems.

Nanoscale, 2021, 13, 8412-8417

Figure R14 | (a) Excitation time dependence of LPL system. (b) Lifetime curves of phosphorescence in PY–CzA system with excitation time for 0 min and 15 min. Both the semi-log plot (left) and log-log plot (right) showed there was no excitation time dependence in dynamic coupling-induced RTP.

Through the above discussion, we could conclude that even though charge transfer excited state/excplex existed both in LPL system and our system, LPL and dynamic coupling-induced phosphorescence underwent very **different photophysical process, mechanism, emission decay kinetics**, and had **different characteristics**.

Secondly, the phosphorescence quantum yields Φ_P in our systems were not very high, primarily due to the intrinsic poor phosphorescence performance of guests themselves. However, our goal was not to select the best candidate to achieve the brightest performance, but rather to **enhance the phosphorescence of molecules that originally exhibited poor ISC efficiency**. Compared to other well-established strategies, such as polymer matrix or amorphous host, Φ_P in our systems had already shown a remarkable enhancement of 135 to 2950 times, highlighting the superior efficiency of dynamic coupling-induced RTP over traditional strategies (see response to Q1 for more details).

Last but not least, although the absolute Φ_P in our system didn't exceed 15%, it is noteworthy that the phosphorescence lifetime τ_P in our system reached up to **2447 ms**, representing the longest red RTP reported to date. Given this record-breaking lifetime and the substantial quantum yield enhancement of up to **2950 times** compared to other well-established systems, dynamic coupling-induced phosphorescence demonstrated its strong strength and emerged as a highly effective strategy for RTP design.

Finally, we would like to thank you once again for your positive feedback. We have revised the manuscript based on your suggestions and comments accordingly, and we hope that these changes adequately address your concerns.

1. It should be demonstrated whether the complex-based approach is more efficient compared to the polymeric host-based and/or crystallization-based approaches for RTP enhancement. To do that, the RTP quantum yields of the most efficient RTP emitter (e.g. PY-*d*₁₀) should be recorded in polymeric host (e.g. PMMA) and/or amorphous host (e.g. PY-*d*₁₀(1 wt. %)-BrCzA). I suggest to shift a part of Supplementary Table 4 to the main text, including into this table the most important photophysical data.

Response: We are very grateful for your insightful suggestions regarding the comparison of dynamic coupling-induced RTP with other well-established strategies. We doped PY, and PY-*d*₁₀ in the polymer matrix, PMMA, and amorphous host, β -estradiol. Both are commonly used matrices in the field of RTP. Their RTP performance was measured and summarized in Table 1.

In PY-PMMA system, phosphorescence was **not observable** at room temperature. Moreover, in the most well-known amorphous host, β -estradiol, phosphorescence could be detected at room temperature, but Φ_P was only 0.002%, accompanied by a short lifetime of 134 ms. Both suggested the intrinsic poor phosphorescence performance of PY. However, in our systems, Φ_P significantly increased to as high as 5.9% (PY-BrCzA), demonstrating a remarkable enhancement of 2950 times higher than β -estradiol system. This highlighted the superior efficiency of dynamic coupling-induced RTP strategy. Similarly, for PY-*d*₁₀ as guest, the Φ_P in dynamic coupling systems showed improvements of 270 to 980 times compared to the PMMA system, and 135 to 490 times when compared to β -estradiol.

Table 1 | Comparison of RTP performance among different host-guest systems.

Sample	System	Φ_P (%) ^a	τ_P (ms)
PY-PMMA	Polymer	/ ^b	178 ^c
PY- β -estradiol	Amorphous steroid	0.002	134
PY-BrCzA		5.9	104
PY-CzBP	Dynamic coupling	0.8	433
PY-BPCzA		1.1	414
PY- d ₁₀ -PMMA	Polymer	0.01	98
PY- d ₁₀ - β -estradiol	Amorphous steroid	0.02	1536
PY- d ₁₀ -BrCzA		9.8	216
PY- d ₁₀ -CzBP	Dynamic coupling	2.7	2447
PY- d ₁₀ -BPCzA		3.3	2268

^a Φ_P was calculated by the phosphorescence component in the photoluminescence spectra. If phosphorescence was too weak to be distinguished from fluorescence in the photoluminescence spectra, Φ_P was determined based on phosphorescence component associated with relevant lifetimes.

^b No phosphorescence was detected at room temperature.

^c Measured at 77 K.

As you suggested, we inserted Table 1 into the main text to emphasize the superior efficiency of dynamic coupling-induced phosphorescence and our changes made to the revised manuscript and Supplementary Information are highlighted in yellow as follows:

On page 11 in the revised manuscript:

Tuning Dynamic Coupling to Enhance RTP Performance

The term ‘dynamic’ implied the coupling process can be further tuned to.....

On page 12 in the revised manuscript:

Thus, after reasonably tuning the dynamic coupling process, RTP performance of PY, a guest with poor intrinsic ISC efficiency, was significantly enhanced, surpassing a series of well-established host–guest systems (Table 1). For instance, in PY–PMMA system, phosphorescence was not observable at room temperature. In the widely recognized amorphous host, β -estradiol²⁵, Φ_P was only 0.002%, accompanied by a short lifetime of 134 ms. However, in dynamic coupling system PY–BrCzA, Φ_P significantly increased to 5.9%, demonstrating a remarkable enhancement of 2950 times higher than β -estradiol system. Similarly, for PY-*d*₁₀ as the guest, the Φ_P in dynamic coupling systems showed improvements of 270 to 980 times compared to the PMMA system, and 135 to 490 times when compared to β -estradiol. This substantial enhancement in quantum yield, along with the record-breaking lifetime ($\tau_{P,max} = 2447$ ms), highlighted the remarkable strength and superior effectiveness of dynamic coupling-induced RTP strategy over traditional design strategies.

On page 14 in the revised manuscript:

Table 1 | Comparison of RTP performance among different host–guest systems.

Sample	System	Φ_P (%) ^a	τ_P (ms)
PY–PMMA	Polymer	γ^b	178 ^c
PY– β -estradiol	Amorphous steroid	0.002	134
PY–BrCzA	Dynamic coupling	5.9	104
PY–CzBP		0.8	433
PY–BPCzA		1.1	414
PY- d ₁₀ –PMMA	Polymer	0.01	98
PY- d ₁₀ – β -estradiol	Amorphous steroid	0.02	1536
PY- d ₁₀ –BrCzA	Dynamic coupling	9.8	216
PY- d ₁₀ –CzBP		2.7	2447
PY- d ₁₀ –BPCzA		3.3	2268

^a Φ_P was calculated by the phosphorescence component in the photoluminescence spectra. If phosphorescence was too weak to be distinguished from fluorescence in the photoluminescence spectra, Φ_P was determined based on phosphorescence component associated with relevant lifetimes.

^b No phosphorescence was detected at room temperature.

^c Measured at 77 K.

2. Supplementary Figure 42 shows the UV-Vis absorption spectrum of the system PY-CzA. This spectrum is characterized by a low-energy band attributed to the complex (but not exciplex formed between PY and CzA) observed in the ground state. To be sure that this band is not related to the RTP emitter PY, the absorption spectrum of the neat film of the emitter PY should be added to Supplementary Figure 42.

Response: Thank you for your constructive suggestions. The absorption spectrum of the neat PMMA film containing 1.0 wt% PY (the same as PY-CzA system) was added to Supplementary Fig. 42. It can be seen the newly observed peak at 375 nm in PY-CzA system was not related to PY.

Supplementary Figure 42 | UV-Vis absorption spectra of PY-PMMA, CzA crystals and PY-CzA crystals.

Our changes made to the revised Supplementary Information are highlighted in yellow as follows:

On page 31 in the revised Supplementary Information:

Supplementary Figure 42 | UV-Vis absorption spectra of PY-PMMA, CzA crystals and PY-CzA crystals.

3. The excitation spectra of the neat films of the host CzA and guest PY should be added to Figure 4a.

Response: Thank you for your valuable suggestion. In Figure 4a, the excitation spectrum of the complex emission, labeled as L_C (total), comprised both dynamic and static components. The dynamic coupling complex was formed through the coupling of the excited-state CzA with the ground-state PY. Therefore, the excitation spectrum of dynamic component should align with that of CzA. Consequently, we used the excitation spectrum of CzA crystals to represent the excitation spectrum of dynamic complex, and marked as $L_{C,D}$ in Figure 4a.

For clarity, we had noted in the caption that the excitation spectrum of the dynamic complex component was obtained by measuring the excitation spectrum of CzA crystals. Also, the excitation spectrum of PY was added to Figure 4a for reference.

Our changes made to the revised manuscript are highlighted in yellow as follows:

On page 9 in the revised manuscript:

Fig. 4 | a, Analysis of excitation spectra of PY-CzA. The excitation spectrum of the complex emission was measured at 470 nm (L_C ; blue solid line). The excitation spectrum of phosphorescence was measured at 595 nm with a delay of 10 ms (P_G ; red solid line). The excitation spectrum of the dynamic complex component was obtained by measuring the excitation spectrum of CzA crystals ($L_{C,D}$; line with blue circles). The excitation spectrum of the static complex component was obtained by subtracting the dynamic component from the excitation spectrum of the total complex emission ($L_{C,S}$; line with green circles). The excitation spectrum of PY-PMMA was attached for reference (gray line). Inset: the difference between dynamic and static complexes.

4. Taking into account that intermolecular charge transfer (exciplex-like) can be observed for ground state [“A simple method to measure intermolecular charge-transfer absorption of organic films.” *Org. Electron.* 2018, 62, 511–515], the differences between exciplexes and the observed complex should be discussed in more detail.

Response: Thank you for your helpful comments. Firstly, we would like to clarify these terms related to the intermolecular charge transfer complex. According to *Modern Molecular Photochemistry of Organic Molecules* by Turro, N. J. et al., intermolecular charge transfer complexes could be categorized into exciplex, and ground-state absorption complex. The coupling complex observed in our work was comprised of both exciplex (dynamic complex), and ground-state absorption complex (static complex).

Secondly, we would like to discuss the difference between exciplex (dynamic complex), and ground-state absorption complex (static complex). As shown in Figure R15, the exciplex was formed through the coupling of one excited-state molecule with one ground-state molecule. Thus, its excitation spectra would align with either the host or guest component. Furthermore, since the coupling occurred only in the excited state and dissociated in the ground state, no new absorption peaks would be observed in the UV-Vis absorption spectrum. In contrast, the ground-state absorption complex coupled in both the excited and ground states, resulting in the appearance of new peaks in the red-shift region of the absorption spectra of both the host and guest components. Since they already coupled in the ground state, this newly emerged peak would also be observed at the same wavelength in the excitation spectrum.

Figure R15 | Difference between exciplex (dynamic complex) and ground-state absorption complex (static complex).

Based on the above discussion, exciplex and ground-state absorption complex could be distinguished through excitation and absorption spectra. Herein, we integrated the excitation analysis (Figure 4a) and absorption analysis (Supplementary Fig. 42) together in Figure R16. In Figure R16a, from 300 nm to 373 nm, the excitation spectrum of complex resembled that of CZA crystals, indicating an exciplex formed through coupling excited-state CZA and ground-state PY. However, a new shoulder peak appeared at approximately 375 nm, coinciding with a newly observed absorption peak in the UV-Vis absorption spectrum (Figure R16b). This suggested the formation of a ground-state absorption complex. Thus, both exciplex and ground-state absorption complex existed in our work.

Figure R16 | Analysis of excitation spectra (a) and absorption spectra (b) for PY–CzA system.

Last but not least, since this work still discussed the host–guest RTP system, we focused on whether the complex could dissociate to the triplet state of the guest, thereby generating RTP. Thus, our attention was on whether the complex could undergo coupling and decoupling, or in other words, whether a dynamic process occurred. Consequently, we referred to the exciplex as dynamic complex and the ground-state absorption complex as static complex in the main text, to emphasize the dynamic coupling process.

As for the work in *Org. Electron.* 2018, 62, 511–515, in Page 1, Paragraph 2 and 3, the author also claimed that no new peak would be observed in UV-Vis absorption spectra for exciplex: ‘*absorption in exciplex-forming molecules has not been detected in the sub-bandgap region by normal reflection–transmission measurements*’. They further noted that measuring the absorption of exciplexes in previous studies required specialized techniques such as ‘*Fourier transform photocurrent spectroscopy (FTPS) and photothermal deflection spectroscopy (PDS) measurements*’. Additionally, they did not observe new absorption peaks in the UV-Vis absorption spectra of their exciplex system, as shown in Page 2, Figure 1c. Instead, they utilized a transfer-matrix method combined with ellipsometry measurements to infer the absorption of the exciplex. Specifically, they compared the experimental and calculated absorption spectra and indirectly fitted the extinction coefficients of the exciplex (Figure 2 and 3), rather than measuring them directly. Thus, their conclusion was consistent with that of Turro, N. J. et al. and supported our viewpoint.

We hope our explanation of the different terms and comments on the paper you mentioned can address your concerns.

5. If static and dynamic complexes are observed (Figure 4a), it is not clear why only dynamic coupling is mentioned in the title. Is the effect of the formation of the static complex negligible in RTP enhancement?

Response: We appreciate your precious comments. As shown in Figure 4e, only dynamic coupling would facilitate the excitons transfer to T_G and promote RTP, while static coupling would trap the excitons and quench RTP. Our key point was dynamic coupling-induced phosphorescence, thus we only mentioned dynamic coupling in the title.

Figure 4 demonstrated the necessity of dynamic coupling in facilitating RTP in detail. Firstly, Figure 4a differentiated the dynamic coupling and static coupling in PY-CzA system through excitation spectra analysis (excitation wavelength from 300nm 373 nm: dynamic; excitation wavelength from 373 to 400 nm: static coupling as discussed in response to Q4). Figure 4b showed that complex emission in dynamic coupling region, $L_{C,D}$, would transfer to P_G at a delay of 10 ms, while that in static coupling region, $L_{C,S}$, would remain at the same position. This proved that only excitons on dynamic coupling complex could be transferred to T_G , while static coupling complex would trap excitons and decay at T_C . Furthermore, the theoretical simulation visualized the dynamic process in the excited state, where the host and guest molecule would firstly couple at S_C and T_C , and then undergo a decoupling process to transfer excitons to T_G . Last but not least, Figure 4d showed that as the decoupling process was suppressed by external pressure, the dynamic coupling was transformed into static coupling, resulting in the quenching of RTP. This further proved the dynamic coupling-induced phosphorescence through *reductio ad absurdum*. Finally, Figure 4e illustrated the different impacts of static and dynamic coupling on RTP through a schematic diagram, providing the audience with a clearer summary. In static coupling, the static complex was already formed in the ground state. Upon excitation to S_C and subsequent ISC to T_C , rather than undergoing significant conformational reconfiguration to a dissociated T_G state, the static complex remained coupled and decayed back to the ground state, exhibiting L_C emission. In dynamic coupling, the dynamic complex was formed after excitation to the excited state. After undergoing the ISC process to T_C , the dynamic complex could dissociate and facilitate the transfer of excitons to T_G . Thus, the excitons decayed from T_G , activating RTP.

Our changes made to the revised manuscript are highlighted in yellow as follows:

On page 9 in the revised manuscript:

Thus, the different impacts of static and dynamic coupling on RTP were summarized in Fig. 4e.

On page 10 in the revised manuscript:

Fig. 4 | e, Schematic diagram for different impact of static (left) and dynamic (right) coupling on RTP (P_G).

6. The previously proposed “dynamic” state-energy diagrams should be mentioned in the introduction [10.1016/j.cej.2020.127902]. In addition, examples of the recent most efficient RTP emitters should be mentioned [10.1039/D3TC04514E or 10.1021/acssuschemeng.3c04011].

Response: Thank you for your careful review and precious suggestions. Our changes made to the revised manuscript are highlighted in yellow as follows:

On page 2 in the revised manuscript:

Recently, the development of pure organic phosphorescent materials has become a hot research topic due to their superior flexibility, processability and biocompatibility¹⁰⁻¹⁸.

Strategies such as incorporation of heavy atoms¹⁹⁻²¹ or (n, π^*) transitions^{12,22,23} to accelerate ISC, crystallization^{13,24} or deuteration²⁵ to suppress non-radiative pathway, are developed to enable room temperature phosphorescence (RTP) in single-component systems.

This static model failed to capture the dynamic nature within the host–guest system, where the continuous molecular motions would affect the intermolecular orbital overlap and reorganize the distribution of excitation energy⁴⁸⁻⁵⁰.

On page 17 in the revised manuscript:

18. Stanitska, M., Volyniuk, D., Minaev, B., Agren, H. & Grazulevicius, J. V. Molecular design, synthesis, properties, and applications of organic triplet emitters exhibiting blue, green, red and white room-temperature phosphorescence. *J. Mater. Chem. C* **12**, 2662–2698 (2024).

19. Volyniuk, L. et al. Single-molecular white emission of organic thianthrene-based luminophores exhibiting efficient fluorescence and room temperature phosphorescence induced by halogen atoms. *ACS Sustainable Chem. Eng.* **11**, 16914–16925 (2023).

On page 18 in the revised manuscript:

48. Andruleviciene, V. et al. TADF versus TTA emission mechanisms in acridan and carbazole-substituted dibenzo[a,c]phenazines: towards triplet harvesting emitters and hosts. *Chem. Eng. J.* **417**, 127902 (2021).

The serial numbers of references were also re-ordered in the revised manuscript.

Reviewer: 3

In this work, Li et al. discovered a dynamic coupling interaction in host-guest doping systems that enhanced room temperature phosphorescence, which was a highly innovative finding. They captured the previously overlooked excited-state coupled intermediate and provided a detailed mechanism investigation on the underlying photophysical process. More importantly, they proposed the different behaviors between dynamic and static coupling in the excited state through theoretical simulations and pressure-dependent experiments, demonstrating the necessity of dynamic process. This mechanism model exhibited broad universality across different systems, and they successfully regulated the dynamic coupling process to further enhance RTP performance. Overall, this work resolved the long-standing issue of unclear mechanisms in host-guest RTP systems, proposed a new mechanism model, and achieved an exceptional record for red RTP. Therefore, I believe that it will attract significant attention in the fields of photophysics and material science, and is worthy of publication in Nature Communications after minor revision.

Response: We would like to thank you for your positive comments and valuable suggestions. We have carefully made a proper revision of the manuscript.

1. What is the design principle for dynamic coupling based RTP system? How to choose suitable host and guest molecules?

Response: Thank you for your valuable comments. In Section IV of the Supplementary Information, we testified the universal applicability of dynamic coupling-induced phosphorescence in 17 different host-guest systems. Through further analyzing these systems, we explained the design principle for choosing suitable host and guest molecules to construct dynamic coupling based RTP system. We are sorry for any unclear description and would like to reorganize our discussion for better understanding.

Firstly, we investigated 17 different host-guest systems and they could be categorized into three scenarios according to their luminescence performance (Supplementary Figs. 49–67):

- (1) If no coupling was observed, the system wouldn't exhibit RTP (Supplementary Figs. 65a and 66a).
- (2) If the energy level of T_C was lower than T_G , the excitons would be trapped by static coupling, and the system wouldn't exhibit RTP (Supplementary Fig. 65b and 66b).
- (3) If dynamic coupling was observed, the system would exhibit RTP (Supplementary Fig. 65c and 66c).

Through analyzing the structure characteristics in three scenarios, we could find that, since the formation of the coupling complex required the molecules to be close to a certain distance, non-planar structures like TPE would hinder this. Additionally, since the coupling complex was stabilized by intermolecular charge transfer process, if this interaction was too weak, there would be no coupling and no RTP (Supplementary Fig. 65a). Conversely, if the interaction was too strong, the complex would be stabilized to a lower energy level than T_G , trapping the exciton within the complex (T_C) and resulting in no RTP (Supplementary Fig. 65b).

Thus, the structure-property relationship/design principle could be deduced as below.

- (1) **Host and guest molecules should have a certain degree of planarity to enable close proximity to one another.**
- (2) **Their electron-donating and accepting tendencies should align within a specific range to form suitably intermolecular charge transfer interactions without being too strong to trap the excitons.**

We then showcased two examples about how to utilize the above design principle to guide us to construct dynamic coupling based RTP and in Supplementary Fig. 67. In systems where excitons were trapped in static coupling, weakening their interactions could activate phosphorescence (Supplementary Fig. 67a). For example, in PZ–BrCzA, PZ had a strong electron-donating tendency, and BrCzA acted as an electron acceptor. They formed an excessively strong complex that trapped the exciton. By weakening the electron-accepting ability of BrCA by replacing with CzA, dynamic coupling was successfully achieved, thus activating phosphorescence in PZ–CzA (Supplementary Fig. 67b). Conversely, in NA–CzA, NA had a strong electron-accepting tendency, and CzA worked as an electron donor. The intermolecular interaction between CzA and NA was too strong and there was no RTP in NA–CzA. By brominating CzA to weaken its electron-donating ability, BrCzA successfully facilitated dynamic coupling and activated the RTP in NA–BrCzA (Supplementary Fig. 67c).

Supplementary Figure 65 | Three scenarios among 17 host–guest systems.

Supplementary Figure 66 | Photophysical processes in above 17 host–guest systems.

Supplementary Figure 67 | Guiding the design of host-guest RTP systems based on structure-property relationship. (a) Schematic diagram for regulation from static to dynamic coupling. (b) Prompt and delayed PL spectra of NA-CzA and NA-BrCzA system. (c) Prompt and delayed PL spectra of PZ-BrCzA and PZ-CzA system.

To sum up, we deduced the design principle for dynamic coupling based RTP system through analyzing 17 different host–guest systems and utilize the principle to guide us design dynamic coupling system with two examples.

We rewrote Section IV in Supplementary Information to give a more detailed description, supplemented two examples about how we utilized the design principle to construct dynamic coupling-induced RTP systems, and added the discussion on design principle to the main text to emphasize its importance. Our changes made to the revised manuscript and Supplementary Information are highlighted in yellow as follows:

On page 11 in the revised manuscript:

Design Principle for Dynamic Coupling

To enrich the emission color and expand the applicable scope.....

.....

Further analysis revealed that the different performances of these host–guest systems could be categorized into three scenarios, through which the design principle for host–guest RTP systems could be clarified (Supplementary Figs. 65 and 66). Considering that the coupling process required close proximity and matching orbital energy levels, while the decoupling process necessitated appropriate intermolecular interactions, the corresponding structure-property relationship for dynamic coupling-induced phosphorescence was deduced as follows: (1) Host and guest molecules should have a certain degree of planarity to enable close proximity to one another; (2) Their electron-donating and accepting tendencies should align within a specific range to form suitably attractive intermolecular charge transfer interactions without being too strong to trap the excitons. Based on this, we further showcased how to utilize such structure-property relationship to guide the design of host–guest RTP systems, and successfully activated RTP through regulating the coupling process in host–guest systems (Supplementary Fig. 67).

On page 53 in the revised Supplementary Information:

Summary of structure-property relationship for host–guest RTP systems

In the first scenario, neither L_C nor P_G was found in these systems, indicating there was no coupling between host and guest molecules to facilitate RTP. The lack of coupling might be attributed to the non-planar structure, which hindered the proximity between the host and guest molecules (TPE), or to the inappropriate intermolecular interactions (BDZ, AN). The ISC processes in host and guest molecules themselves were inefficient. The TTET process from T_H to T_G was also inhibited due to the weak ISC ability of host and the strong non-radiative dissipation from T_H . Thus, no RTP was generated and only $F_{H/G}$ was observed (Supplementary Figs. 65a and 66a).

In the second scenario, L_C appeared in the PL spectra, but P_G was not generated. This indicated coupling occurred but the ET pathway to T_G during decoupling process was inhibited. The reason was that the intermolecular interactions between host and guest molecules were too strong, forming static complexes with T_C lower than T_G . This made the ET process to T_G energetically unfavored.

Thus, the excitons were trapped within the static complex and decayed from T_C (Supplementary Figs. 65b and 66b).

In the third scenario, both L_C and P_G were observed in these systems, indicating the excitons were successfully transferred to T_G state by dynamic coupling process. This highlighted the essential role of dynamic coupling in facilitating RTP. In these systems, both the host and guest molecules exhibited planar structures. Furthermore, their electron-donating abilities and accepting tendencies matched well with each other, leading to attractive intermolecular interactions that allowed them to couple effectively without being overly strong. Both factors benefited the formation of dynamic coupling (Supplementary Figs. 65c and 66c).

The above discussion clarified the structure-property relationship for dynamic coupling-induced phosphorescence:

(1) Host and guest molecules should have a certain degree of planarity to enable close proximity to one another.

(2) Their electron-donating and accepting tendencies should align within a specific range to form suitably intermolecular charge transfer interactions without being too strong to trap the excitons.

This structure-property relationship could guide us to design host–guest RTP systems. For example, in systems where excitons were trapped in complex, weakening their interactions could activate phosphorescence (Supplementary Fig. 67a). In PZ–BrCzA, PZ had a strong electron-donating tendency, and BrCzA acted as an electron acceptor. They formed an excessively strong complex that trapped the exciton. By weakening the electron-accepting ability of BrCA by replacing with CzA, dynamic coupling was successfully achieved, thus activating phosphorescence in PZ–CzA (Supplementary Fig. 67b). Conversely, in NA–CzA, NA had a strong electron-accepting tendency, and CzA worked as an electron donor. The intermolecular interaction between CzA and NA was too strong and there was no RTP in NA–CzA. By brominating CzA to weaken its electron-donating ability, BrCzA successfully facilitated dynamic coupling and activated the RTP in NA–BrCzA (Supplementary Fig. 67c).

Supplementary Figure 67 | Guiding the design of host-guest RTP systems based on structure-property relationship. (a) Schematic diagram for regulation from static to dynamic coupling. (b) Prompt and delayed PL spectra of NA-CzA and NA-BrCzA system. (c) Prompt and delayed PL spectra of PZ-BrCzA and PZ-CzA system.

2. What is the quantum yield of the host molecules used in the present system, and how does it change after doping guest molecules?

Response: Thank you for your interest in our host molecules. The photoluminescence quantum yields of host molecules before and after doping PY were measured and summarized in Table R1. All host molecules showed negligible RTP. Except CzA, host molecules exhibited low Φ_{PL} due to their intense non-radiative decay. After doping, they all increased, due to the combined effects of guest fluorescence, complex luminescence, and phosphorescence enhancement. The reason for the reduction in CzA might be due to the enhanced ISC process in host–guest system weakened the fluorescence of CzA.

Table R1 | Photoluminescence quantum yields of host molecules before and after doping PY.

Host	Φ_{PL} (%)	Host–guest system	Φ_{PL} (%)
CzA	42.8	PY–CzA	28.3
BrCzA	0.1	PY–BrCzA	8.2
CzBP	0.6	PY–CzBP	1.2
BPCzA	1.2	PY–BPCzA	1.9

Since Φ_{PL} of host–guest systems were recorded in Table S4, we added Φ_{PL} of host molecules in Supplementary Fig. 32. Our changes made to the revised Supplementary Information are highlighted in yellow as follows:

On page 23 in the revised Supplementary Information:

Supplementary Figure 32 | Prompt and delayed PL spectra of host molecules in the crystal state, CzA (a), BrCzA (b), CzBP (c), BPCzA (d). Delayed spectra were measured at 77 K with a delay time of 10 ms. All host molecules showed negligible RTP at room temperature. Their photoluminescence quantum yields measured at room temperature was noted.

3. The phosphorescence spectra of PZ, NA, BDZ, AN used in guest engineering should be supplemented to confirm the origin of the RTP.

Response: Thank you for your constructive suggestions. The phosphorescence spectra of PZ, NA, BDZ, AN were supplemented.

Our changes made to the revised manuscript and Supplementary Information are highlighted in yellow as follows:

On page 11 in the revised manuscript:

By comparing with the phosphorescence spectra of guest molecules, it was confirmed that these RTP originated from the triplet states of the guests (Supplementary Fig. 51).

On page 39 in the revised Supplementary Information:

Supplementary Figure 51 | Prompt and delayed PL spectra of guest molecules in PMMA film. The delayed spectra were measured at 77 K. Delay time: 10 ms.

4. For host engineering, why is the lifetime of PY-BPCzA shorter than PY-CzBP?

Response: Thank you for your valuable comments. The reason for shorter phosphorescence lifetime of PY-BPCzA was because both the radiative decay and non-radiative decay from T_G state in PY-BPCzA was faster than those in PY-CzBP.

Radiative decay (Figure R17): BPCzA had two carbonyl groups, providing more (n, π^*) than CzBP. Thus, it would accelerate the ISC process. As supported by theoretical calculation, PY-BPCzA had more efficient ISC process than PY-CzBP. Thus, the SOC between T_1 and S_0 would also be enhanced, leading to a faster radiative decay from T_G . As shown in Figure R17c, the experimental rate constant of radiative decay in PY-BPCzA, k_p , was larger than that in PY-CzBP.

Figure R17 | Comparison of radiative decay in PY-CzBP and PY-BPCzA systems. (a) Structure characteristics of CzBP and BPCzA. (b) Calculated energy levels, ISC channels, SOC constants. (c) Experimental photophysical properties.

Non-radiative decay (Figure R18): The bulky hexyl group and two benzoyl groups in BPCzA occupied larger space around the molecule, which would form a larger steric hindrance and weaken the intermolecular interactions, resulting in a looser molecular packing in PY-BPCzA than that in PY-CzBP. Theoretical calculation also showed that the intermolecular distance within PY-BPCzA complex was larger than PY-CzBP, suggesting more non-radiative dissipation of triplet excitons in PY-BPCzA. As expected, the experimental rate constant of non-radiative decay, k_{nr} , in PY-BPCzA was larger than that in PY-CzBP.

Figure R18 | Comparison of non-radiative decay in PY-CzBP and PY-BPCzA systems. (a) Structure characteristics of CzBP and BPCzA. (b) Calculated intermolecular distance within complexes. The intermolecular distance, d_{avg} , was determined as the average distance between the 1, 3, 6, 8, and 9 positions of carbazole moiety and the plane of pyrene. (c) Experimental photophysical properties.

5. In Figure 6f, achieving full-color RTP in one sample is very interesting, but the term 'sunlight' should be added below the first photo for clarification.

Response: We are grateful for your careful review and suggestions. The term 'sunlight' was added in Figure 6f.

Our changes made to the revised manuscript are highlighted in yellow as follows:

On page 15 in the revised manuscript:

Fig. 6 | Applications for 4D encryption (a-c) and full-color display (d-f), f. Photos of PY-BPCzA crystals captured after ceasing 365 nm excitation source at different temperature. The photo of PY-BPCzA crystals under sunlight was also shown.

We appreciate all the suggestions and comments of reviewers and supplemented a lot of experiments and calculations. Considering that some new discussion, Supplementary Figures and references are added during the revision, the serial numbers of pages, Supplementary Figures, and references are also re-ordered in the revised manuscript and Supporting Information with double-check.

Summary of the reviewers' comments:

Reviewer #1 (Remarks to the Author):

In the revised manuscript, the authors have implemented certain modifications and provided clarifications in response to the feedback, resulting in a noticeable enhancement in the overall quality of the work. Nevertheless, I continue to find the responses to comments 2 and 4 unsatisfactory, particularly with respect to the construction of the host-guest interaction model, which I consider to be fundamentally flawed. In the absence of a precise stacking model, the computational data presented lacks credibility. I strongly advise the authors to remove the associated data from the manuscript. Additionally, regarding the dynamic coupling interactions between the host and guest molecules, I recommend that the authors explore the development of empirical or semi-empirical physical parameters to quantitatively characterize the strength of these interactions, as opposed to relying on a macroscopic and qualitative assessment. Therefore, I still maintain my stance that the manuscript, in its present form, is not suitable for publication.

Reviewer #2 (Remarks to the Author):

Our comments are logically addressed. We suggest to accept the manuscript.

Reviewer #3 (Remarks to the Author):

The authors have well addressed my concerns. I think the quality of the manuscript is significantly improved. Thus, I recommend the acceptance of the manuscript by the journal as is.

Point-by-point response to the reviewers' comments:

(Reviewers' comments and suggestions: in black; Responses to the comments and suggestions: in blue)

Reviewer #1 (Remarks to the Author):

In the revised manuscript, the authors have implemented certain modifications and provided clarifications in response to the feedback, resulting in a noticeable enhancement in the overall quality of the work. Nevertheless, I continue to find the responses to comments 2 and 4 unsatisfactory, particularly with respect to the construction of the host-guest interaction model, which I consider to be fundamentally flawed. In the absence of a precise stacking model, the computational data presented lacks credibility. I strongly advise the authors to remove the associated data from the manuscript. Additionally, regarding the dynamic coupling interactions between the host and guest molecules, I recommend that the authors explore the development of empirical or semi-empirical physical parameters to quantitatively characterize the strength of these interactions, as opposed to relying on a macroscopic and qualitative assessment. Therefore, I still maintain my stance that the manuscript, in its present form, is not suitable for publication.

Response: We sincerely appreciate the reviewer's insightful comments and constructive suggestions, which have greatly helped us improve the quality of our manuscript. Thank you for your recognition for the improvement of our revised manuscript. We deeply regret that our previous responses to comments 2 and 4 did not fully address your concerns. In this revised version, we have carefully followed your suggestions by removing the debatable computational model section and supplementing the discussion with additional experimental data. Furthermore, as suggested, we have developed semi-empirical and empirical physical parameters to quantitatively characterize the strength of dynamic coupling interactions under different situations. Once again, we are grateful for the time and effort you have dedicated to reviewing our work. Your feedback has been invaluable, and we hope that these revisions adequately address your concerns.

(1) In the absence of a precise stacking model, the computational data presented lacks credibility. I strongly advise the authors to remove the associated data from the manuscript.

Response: We sincerely thank the reviewer for your precious suggestions and insightful critique regarding the limitations of our theoretical simulations. We fully acknowledge your concerns about the reliability of computational data in reflecting the actual conformation of complex in our system. Despite extensive efforts—including cocrystal cultivation, single-crystal X-ray diffraction analysis, and PXRD studies of doped samples—we were unable to experimentally determine the precise stacking model between the host and guest molecules. In light of your valid criticism, we agree that the computational results lack sufficient reliability and have therefore removed them from the main text, adjusted the relevant discussion sections accordingly, and revised Figure 4 by supplementing more experimental evidence.

(2) Additionally, regarding the dynamic coupling interactions between the host and guest molecules, I recommend that the authors explore the development of empirical or semi-empirical physical

parameters to quantitatively characterize the strength of these interactions, as opposed to relying on a macroscopic and qualitative assessment.

Response: We sincerely appreciate your valuable suggestions, which prompted us to develop some semi-empirical physical models based on experimental data to provide a more quantitative description of host–guest interactions in the dynamic coupling process.

I. Overall description of physical parameter: ΔG_{dc}^\ddagger

First, we briefly revisit that the coupled complex forms through the intermolecular charge-transfer (CT) process between host and guest molecules, generating CT states (S_C and T_C) with enhanced ISC efficiency. Whether RTP is turned on depends on whether the coupling complex can undergo a decoupling process, enabling the excitons to transition from the T_C state to the triplet state of guest, T_G (Figure R1).

Figure R1. Illustration of dynamic coupling-induced RTP.

In the T_C state, electrons are delocalized across the entire complex, exhibiting CT characteristics, whereas in the T_G state, electrons are localized on the guest molecule, displaying locally excited (LE) characteristics. Thus, the decoupling process (i.e. T_C to T_G transition) inherently involves an electron exchange process to revert the CT state to the LE state. According to the Arrhenius equation (Eq. S1), the rate constant of this electron exchange process, k_{ex} , is governed by the activation energy barrier of decoupling process, ΔG_{dc}^\ddagger .

$$k_{ex} = k_0 \exp\left(-\frac{\Delta G_{dc}^\ddagger}{RT}\right) \quad \text{Eq. S1}$$

k_{ex} , rate constant of electron exchange during decoupling process; k_0 , pre-exponential factor, representing the maximum k_{ex} when ΔG_{dc}^\ddagger is 0 kcal/mol. ΔG_{dc}^\ddagger is the activation energy for decoupling process.

Consequently, we propose the activation energy ΔG_{dc}^\ddagger as a semi-empirical parameter to quantify the dynamic coupling interaction and correlate it with RTP performance (Figure R2). A higher activation energy corresponds to a slower electron exchange rate, thereby suppressing the decoupling process and weakening RTP emission from the T_G state, and *vice versa*.

Figure R2. Illustration of the relationship between ΔG_{dc}^\ddagger and RTP performance.

Below we will elaborate in detail on the application of this parameter to quantify dynamic coupling interactions in the host–guest systems investigated in our study. Specifically, we systematically examine the mathematical representations of this physical parameter in two scenarios:

- (i) *Comparison of ΔG_{dc}^\ddagger across different host–guest systems* (analyzing electronic effect based on Marcus theory, Section II).
- (ii) *Modulation of ΔG_{dc}^\ddagger within identical host–guest system* (evaluating proximity effect based on Coulomb's law, Section III).

II. Comparison of ΔG_{dc}^\ddagger across different host–guest systems: electronic effect

II.1 Mathematical representation of ΔG_{dc}^\ddagger

As discussed in Section I, given that the decoupling process intrinsically involves electron exchange, the relative electron-donating and electron-accepting capabilities of the host and guest components will influence the activation energy barrier and ultimately govern the decoupling propensity of the coupling complex. According to Marcus theory¹, ΔG_{dc}^\ddagger is associated with the thermodynamic free energy difference, ΔG_{dc}^0 , and reorganization energy, λ , during decoupling process (Eq. S2).

$$\Delta G_{dc}^\ddagger = \frac{(\Delta G_{dc}^0 + \lambda)^2}{4\lambda} \quad \text{Eq. S2}$$

ΔG_{dc}^0 , representing the energy difference between coupling complex and decoupled host/guest pairs after electron exchange, can be estimated by measuring the electrochemical potentials for the oxidations $E_{(D^+/D)}^0$ and reductions $E_{(A/A^-)}^0$ via cyclic voltammetry experiments¹, as per Eq. S3.

The organizational energy (λ) of the system, representing internal molecular reorganization from T_C state to T_G state, since external solvent reorganization is negligible in the crystal state, can be estimated using DFT².

$$\Delta G_{dc}^0 \approx \mathcal{F}E_{(D^+/D)}^0 - \mathcal{F}E_{(A/A^-)}^0 - E_{D^*/A^*} \quad \text{Eq. S3}$$

E_{D^*/A^*} denotes the excitation energy of either the donor or acceptor, depending on which molecular entity is actually excited in the photophysical process.

Based on Eqs. S2–S3, we can determine the semi-empirical physical parameter ΔG_{dc}^\ddagger for each system by calculating the corresponding ΔG_{dc}^0 and λ values. Thus, we select four representative systems in our work with PY as guest and carbazole derivatives as hosts (PY–CzA, PY–BrCzA, PY–CzBP, and PY–BPCzA). To strengthen our dataset, we incorporate additional PY–BP system where benzophenone (BP) serves as the host. This is because BP represents the fundamental

structural unit of CzBP/BPCzA, and PY-BP has been widely reported as an efficient RTP system³⁻⁵. Their ΔG_{dc}^0 , λ and ΔG_{dc}^\ddagger values are listed in Table R1. Supporting cyclic voltammetry curves used for determining the electrochemical potentials are provided in Figure R3 and Table R2.

Table R1. Thermodynamic parameters of host-guest systems during dynamic coupling process.

	PY-CzA	PY-BrCzA	PY-CzBP	PY-BPCzA	PY-BP
ΔG_{dc}^0 (kcal/mol)	-36.50	-38.28	-20.24	-34.55	-50.73
λ (kcal/mol)	7.40	10.67	15.27	16.81	8.81
ΔG_{dc}^\ddagger (kcal/mol)	28.60	17.85	0.40	4.69	49.88
τ_P (ms)	363	104	433	414	315

Figure R3. Cyclic voltammetry curves of PY and host molecules in MeCN.

Table R2. The electrochemical potential for the oxidation and reduction of PY and host molecules.

	PY	CzA	BrCzA	CzBP	BPCzA	BP
$E_{(M^+/M)}^0$ (V)	1.20	1.10	0.80	1.27	1.49	2.10
$E_{(M/M^-)}^0$ (V)	-1.93	-2.20	-2.07	-1.55	-1.50	-1.65

II.2 Correlation of ΔG_{dc}^\ddagger with RTP performance

As established in Section I, the rate constant of electron exchange during decoupling process (k_{ex}) directly controls the population of excitons reaching T_G (Φ_T). For systems exhibiting comparable triplet decay rates ($k_P + k_{nr, P}$), the observed phosphorescence lifetime (τ_P) demonstrates a linear dependence on both Φ_T , and consequently, on k_{ex} .

$$\tau_P \propto \Phi_T \propto k_{ex} \quad \text{Eq. S4}$$

Therefore, we selected τ_P as the key performance metric for RTP, while ΔG_{dc}^\ddagger serves as our semi-empirical parameter to quantify the decoupling tendency in the dynamic coupling process.

Based on Eqs. S1 and S4, we can derive an exponential relationship between τ_P and ΔG_{dc}^\ddagger :

$$\tau_P \propto \exp(-\Delta G_{dc}^\ddagger)$$

Consequently, a negative linear correlation exists between $\ln(\tau_P)$ and ΔG_{dc}^\ddagger :

$$\ln(\tau_P) \propto -\Delta G_{dc}^\ddagger$$

Figure R4 reveals an excellent linear correlation ($R^2 = 0.99$) between $\ln(\tau_P)$ and ΔG_{dc}^\ddagger , which strongly supports the proposed mechanism of dynamic coupling-induced RTP. The observed negative slope reveals a systematic trend across the series from PY-CzBP to PY-BPCzA to PY-CzA to PY-BP: as the activation energy increases, the decoupling efficiency progressively decreases, resulting in corresponding reduction of RTP lifetimes, which is well consistent with our discussion in Figure R2. For the PY-BrCzA system, the heavy-atom effect (HAE) significantly increases the triplet decay rates ($k_P + k_{nr, P}$), which makes the proportional relationship between τ_P and k_{ex} in Eq. S4 invalid. Thus, this deviation can be excluded reasonably from the linear fitting analysis (Figure R5). Such an excellent linear correlation observed in Figure R4 strongly suggests that the phosphorescence performance in different host-guest systems can be quantitatively predicted by calculating the activation energy ΔG_{dc}^\ddagger .

Figure R4. Relationship between phosphorescence lifetime τ_P and the activation energy for decoupling process, ΔG_{dc}^\ddagger , for various host-guest systems.

Figure R5. (a) Deviation of PY–BrCzA system from fitted linear relationship due to heavy atom effect. (b) Experimental rate constants of radiative decay from T_G state, k_P , and nonradiative decay from T_G state, $k_{nr,p}$ in these systems.

III. Modulation of ΔG_{dc}^\ddagger within identical host–guest system: proximity effect

Section II systematically examined how electronic effects of host and guest will affect the activation energy ΔG_{dc}^\ddagger of the decoupling process, thereby governing RTP performance across different systems. We now turn to proximity effects within identical host–guest system, where intermolecular distance variation serves as a critical control parameter. By exerting external pressure, we demonstrate how proximity effect will modulate ΔG_{dc}^\ddagger and ultimately influence RTP performance through tuning Coulombic interaction.

III.1 Mathematical representation of ΔG_{dc}^\ddagger

For a certain host–guest system, where the electronic effects of both host and guest molecule are fixed, the variables ΔG_{dc}^0 and λ in Eq. S2 become constants. Thus, we require an alternative mathematical representation to describe the activation energy ΔG_{dc}^\ddagger . Since the complex is formed through coupling between partially charged donor and acceptor moieties, whose relative motion is constrained by Coulombic interactions. In this context, the activation energy required for decoupling corresponds to the energy barrier needed to overcome the Coulombic potential. According to Coulomb’s law, this interaction energy (U) can be quantitatively described by Eq. S5.

$$U = \frac{1}{4\pi\epsilon_0\epsilon_r} \cdot \frac{q_1q_2}{r} \quad \text{Eq. S5}$$

ϵ_0 is the vacuum permittivity and ϵ_r is the relative permittivity. q_1 and q_2 represent the charges carried by the host and guest respectively. Since the host and guest are fixed in one identical system, their electron-donating/accepting capabilities remain constant, which means the quantities of partial charges they carried are fixed. Therefore, all these terms, ϵ_0 , ϵ_r , q_1 , and q_2 can be grouped into constant factors. r represents the intermolecular distance. Since the volume V of a solid scales with r^3 ($V \propto r^3$), and volume is inversely proportional to pressure p ($V \propto 1/p$), we can establish the relationship between U and p as shown in Eqs. S6–S7. Eq. S7 demonstrates that by varying external pressure, we can control the intermolecular distance and thereby modulate the activation energy for decoupling (ΔG_{dc}^\ddagger).

$$\frac{1}{r} \propto p^{1/3} \quad \text{Eq. S6}$$

$$\Delta G_{dc}^{\ddagger} = U = k_c \cdot p^{1/3} \quad \text{Eq. S7}$$

where k_c is a constant term determined by the intrinsic electronic properties of host and guest molecules, as well as the compressibility of the host crystals.

III.2 Correlation of ΔG_{dc}^{\ddagger} with RTP performance

We take PY–BrCzA as an example and applied external pressure in the range of 0-10 GPa to modulate the activation energy by compressing intermolecular distances. Figure R6a shows its PXRD patterns under different pressure. All diffraction peaks shift toward higher angles with increasing pressure, indicating significant reduction in unit cell volume and interplanar spacing. Under 0-6 GPa external pressure, the main crystal diffraction peaks remain observable, demonstrating its crystal structure remain intact up to 6 GPa. At 10 GPa, the diffraction peaks weaken to near undetectable levels, indicating transition from crystalline to amorphous state. Thus, we can confirm that within the 0-6 GPa range, the proportional relationship between $1/r$ and $p^{1/3}$ in Eq. S6 is still valid, and consequently the correlation between ΔG_{dc}^{\ddagger} and $p^{1/3}$ in Eq. S7 remains effective.

Figure R6. (a) PXRD patterns of PY–BrCzA under different pressure. (b) PL spectra of PY–BrCzA under different pressure.

Figure R6b demonstrates the spectral evolution under pressure, where the complex emission (Lc, around 470 nm) gradually weakens and redshifts. This indicates enhanced CT degree as intermolecular distances decrease, implying an increased activation energy barrier for decoupling. As expected, the phosphorescence intensity diminishes. Due to technical limitations in measuring pressure-dependent phosphorescence lifetimes, we utilize the phosphorescence intensity (I_p) as the performance metric. According to Eq. S1, we can derive an exponential relationship between I_p and ΔG_{dc}^{\ddagger} .

$$I_p \propto \exp(-\Delta G_{dc}^{\ddagger})$$

Combining with Eq. S7, a negative linear correlation should exist between $\ln(I_p)$ and $p^{1/3}$.

$$\ln(I_p) \propto -p^{1/3}$$

As demonstrated by the experimental data, Figure R7 shows an exceptionally strong linear correlation ($R^2 = 0.99$) between $\ln(I_p)$ and $p^{1/3}$. Given the established relationship between $p^{1/3}$ and ΔG_{dc}^{\ddagger} in Eq. S7, this equivalently confirms a negative linear dependence of RTP intensity on the activation energy for decoupling (ΔG_{dc}^{\ddagger}). These results provide compelling experimental

validation for our proposed mechanism: increasing external pressure elevates the decoupling activation energy through enhanced Coulombic interactions, thereby diminishing the RTP performance.

Figure R7. Relationship between phosphorescence intensity I_p and the activation energy for decoupling process, ΔG_{dc}^{\ddagger} , in PY–BrCzA system under different pressure.

III.3 Supplementary evidence on enhanced intermolecular coupling under high pressure.

To further demonstrate that the intermolecular coupling is enhanced under high pressure (beyond the PL spectra and PXRD patterns in Figure R6), we conducted additional *in situ* high-pressure characterizations including UV-Vis absorption spectra, time-resolved luminescence decay curves, FT-IR spectra and Raman spectra.

III.3.1 UV-Vis absorption spectra

Figure R8 demonstrates that the absorption spectra of PY–BrCzA crystals exhibit a progressive bathochromic shift in the onset position from 398 nm to 475 nm under increasing pressure. This observation clearly indicates that the electronic coupling between PY and BrCzA is enhanced through compressed intermolecular distance, thereby gradually lowering the energy gap of CT complex state. These spectral changes correlate well with the redshift of complex emission (L_C) observed in Figure R6. The parallel bathochromic trends in both absorption and emission spectra unambiguously provide evidence for the enhanced CT interactions between PY and BrCzA. Upon releasing the pressure back to ambient conditions, the absorption spectrum fully recovered to its original state, suggesting that this pressure-enhanced coupling process is reversible.

Figure R8. UV-Vis absorption spectra of PY-BrCzA under different pressure.

III.3.2 Time-resolved luminescence decay curves

To further investigate the evolution of complex emission (L_C), we measure its lifetime under varying pressure conditions. As the external pressure increases from 0 GPa to 10 GPa, the lifetime of L_C exhibits a significant decrease from 33 ns to 3 ns (Figure R9 and Table R3). Detailed analysis of the radiative (k_r) and non-radiative (k_{nr}) decay rate constants reveals a consistent trend: k_r progressively decreases while k_{nr} keeps increasing with pressure.

This phenomenon can be attributed to two primary factors: First, the enhanced CT character reduces the orbital overlap between HOMO and LUMO, leading to a more forbidden transition from S_C state (singlet state of coupling complex), thereby decreasing k_r . Second, the reduction in intermolecular distance between PY and BrCzA under pressure promotes molecular collisions and facilitates nonradiative energy transfer, resulting in the observed increase in k_{nr} .

Thus, these lifetime measurements also strongly support our conclusion that increasing pressure enhances intermolecular coupling.

Figure R9. (a) Lifetime curves of PY-BrCzA recorded at 500 nm under different pressure. (b) Rate

constants of radiative and non-radiative decay from S_C state within PY–BrCzA complex under different pressure.

Table R3. Photophysical properties of PY–BrCzA under different pressure.

External pressure (GPa)	τ (ns) ^a	k_r ($\times 10^5$ s ⁻¹)	k_{nr} ($\times 10^7$ s ⁻¹)
0.0	32.55	7.07	2.82
0.4	37.19	3.93	2.55
1.1	29.85	2.76	3.25
1.8	24.53	2.66	3.98
4.1	19.01	2.37	5.18
6.1	12.79	1.94	7.75
10.0	3.26	1.92	30.60

^a Lifetime was measured at 500 nm.

III.3.3 FT-IR spectra and Raman spectra

In situ FT-IR and Raman spectra under high pressure were also measured (Figures R10–R12). The single-crystal structure of BrCzA reveals distinct C–H \cdots Br interaction (Figure R10), which can be used as a sensitive indicator for monitoring the evolution of intermolecular interactions under pressure.

As illustrated in Figure R11, the majority of FT-IR absorption peaks exhibit blue shifts with increasing pressure, consistent with bond shortening induced by lattice compression. Some typical absorption bands are assigned. For example, the band at ~ 1280 cm⁻¹ corresponds to in-plane ring-stretching vibrations of the benzene moiety. Its progressive splitting and broadening under pressure directly reflect enhanced π – π stacking interactions. Characteristic stretching modes of C–H and C–Br bonds (annotated in Figure R10) display contrasting spectral trends: a continuous blue shift for $\nu(\text{C–H})$ from 2931 to 2983 cm⁻¹, versus a red shift for $\nu(\text{C–Br})$ from 734 to 723 cm⁻¹. This provides compelling evidence for the strengthening of C–H \cdots Br interactions under high pressure.

Raman spectral evolution corroborates these findings (Figure R12). The C–H bending mode shows a monotonic blue shift, whereas the C–Br bending mode undergoes an initial red shift followed by a blue shift at higher pressures. This might be attributed to the intensified intermolecular interaction dominates at lower pressures, while the bond compression prevail at higher pressure.

Thus, the combined evidence from FT-IR spectra, Raman spectra, complemented by the PXRD analysis in Section III.2, conclusively confirms three key structural changes within PY–BrCzA under high pressure: lattice contraction, shortened intermolecular spacing and amplified intermolecular interactions. These phenomena provide corroborative support for the reinforcement of coupling interaction within the system.

Figure R10. C–H···Br interaction in BrCzA crystal.

Figure R11. FT-IR spectra of PY–BrCzA under different pressure.

Figure R12. Raman spectra of PY-BrCzA under different pressure.

VI. Summary

We have developed the activation energy for decoupling process ΔG_{dc}^\ddagger as a semi-empirical parameter to describe dynamic coupling strength. We systematically discuss the mathematical expressions of ΔG_{dc}^\ddagger under different situation (comparison across different host-guest systems, and modulation within the same system), elucidating the electronic effect and proximity effect during the dynamic coupling process, respectively. Moreover, we established a quantitative relationship between ΔG_{dc}^\ddagger and RTP performance. The strong linear correlation ($R^2 = 0.99$) demonstrates the reliability of using decoupling activation energy to predict RTP performance, shedding light on the design principles for host-guest systems.

Furthermore, in the section discussing proximity effects (Section III), to provide more evidence for enhanced coupling interactions with increasing pressure, we supplemented additional experiments including *in situ* high-pressure UV-Vis absorption spectra, lifetime measurements, FT-IR spectra, and Raman spectra. We hope this will better address your dissatisfaction with our previous response to Q4.

We sincerely appreciate the time and effort you have devoted during our review process. Your invaluable suggestions have transformed our work from *a qualitative description* relying on theoretical simulations to *a solid quantitative study* supported by extensive experimental evidence. Your constructive criticism has enabled us to establish precise quantitative relationships and construct a rigorous physical model. We hope our current improvements can resolve your concerns.

V. Reference

(1) Turro, N. J., Ramamurthy, V. & Scaiano, J. C. *Modern Molecular Photochemistry of Organic Molecules*. (University Science Books, New York, 2010).

- (2) Ma, L. et al. Triplet exciplex mediated multi-color ultra-long afterglow materials. *Angew. Chem. Int. Ed.* **64**, e202500847 (2025).
- (3) Xiao, F. et al. Guest-host doped strategy for constructing ultralong-lifetime near-infrared organic phosphorescence materials for bioimaging. *Nat. Commun.* **13**, 186 (2022).
- (4) Qiu, W. et al. Achieving purely organic room-temperature phosphorescence mediated by a host–guest charge transfer state. *J. Phys. Chem. Lett.* **12**, 4600–4608 (2021).
- (5) Yang, G. et al. Construction and application of large stokes-shift organic room temperature phosphorescence materials by intermolecular charge transfer. *J. Phys. Chem. Lett.* **14**, 6927–6934 (2023).

Our changes made to the revised manuscript and Supplementary Information are highlighted in yellow as follows:

On page 8 in the revised manuscript:

Since the difference between dynamic and static complexes lay in whether they remained coupled during the decay to the ground state, it could be speculated that a similar dissociation process might be involved in the excited-state transition from T_C to T_G (Supplementary Figs. 43 and 44). To verify this, pressure-dependent experiments were conducted to regulate the coupling strength. PY-BrCzA was prepared by doping PY into brominated CzA to enhance the P_G intensity through heavy atom effect, enabling the simultaneous observation of L_C and P_G (Supplementary Figs. 45 and 46). *In situ* pressure-dependent PL spectra (Fig. 4c) shows the emission (L_C) gradually red-shifted with weaker intensity as the pressure increased. This clearly demonstrated the progressively enhanced CT character of complex state, indicating that the intermolecular coupling was strengthened. However, contrary to the anticipated enhancement due to the suppression of molecular vibrations under high pressure^{1,2}, P_G weakened and eventually disappeared. This abnormal phenomenon might be because the decoupling of complex was also constrained by the reduction in the free molecular volume under high pressure.

More structural characterizations and quantitative analysis were carried out. X-ray diffraction showed that all diffraction peaks shift toward higher angles with increasing pressure, indicating significant reduction in unit cell volume and interplanar spacing (Fig. 4d and Supplementary Fig. 47). Taking (004) crystal plane as an example, we calculated its interplanar spacing, which showed a continuous decreasing from 4.56 Å to 4.33 Å. Fig. 4e displayed the variation in the intensities of L_C and P_G , and (004) interplanar spacing under different pressure. Surprisingly, the trend of weakening intensities of L_C and P_G was highly consistent with the trend of reduced interplanar spacing (Supplementary Figs. 48 and 49). Quantitative analysis further investigated the relationship between activation energy during the decoupling process and RTP performance under various pressure (Supplementary Fig. 50). A robust linear correlation ($R^2 = 0.99$) provided compelling experimental validation for our proposed mechanism: shortening the intermolecular distance within complex enhanced the coupling strength and elevated the decoupling activation energy, thereby diminishing the RTP performance.

Furthermore, we observed a continuous bathochromic shift in UV-Vis absorption spectra and progressively shortened lifetimes of L_C with increasing pressure (Supplementary Figs. 51 and 52), enabling calculation of the radiative (k_r) and non-radiative (k_{nr}) decay rate constants (Fig. 4f and Supplementary Table 5). The decrease in k_r originated from diminished HOMO-LUMO orbital overlap resulting from enhanced CT character, while the increase in k_{nr} was due to that the shortened intermolecular distance during the pressurization process intensified collisions and nonradiative energy transfer. Complementary FT-IR and Raman spectra further confirmed the pressure-enhanced intermolecular interactions (Supplementary Figs. 53–55). Together, these experiments demonstrate how external pressure can precisely modulate the coupling strength within the PY-BrCzA system, providing compelling evidence that validated the necessity of dynamic coupling in generating RTP through *reductio ad absurdum*.

On page 10 in the revised manuscript:

Fig. 4 | Demonstration for necessity of dynamic coupling. **a**, Analysis of excitation spectra of PY-CzA. The excitation spectrum of the complex emission was measured at 470 nm (L_C ; blue solid line). The excitation spectrum of phosphorescence was measured at 595 nm with a delay of 10 ms (P_G ; red solid line). The excitation spectrum of the dynamic complex component was obtained by measuring the excitation spectrum of CzA crystals ($L_{C,D}$; line with blue circles). The excitation spectrum of the static complex component was obtained by subtracting the dynamic component from the excitation spectrum of the total complex emission ($L_{C,S}$; line with green circles). The excitation spectrum of PY-PMMA was attached for reference (gray line). Inset: the difference between dynamic and static complexes. **b**, Excitation-phosphorescence mapping of PY-CzA with a delay of 1 (left) and 10 ms (right). The upper insets: the delayed PL spectra excited at 360 nm and 380 nm. P_G appeared through dynamic coupling when PY-CzA was excited at 360 nm but disappeared when static complex was excited at 380 nm. **c**, Pressure-dependent PL spectra of PY-BrCzA excited at 355 nm. **d**, Pressure-dependent X-ray diffraction patterns of PY-BrCzA crystals. The diffraction peak of (004) crystal plane was denoted. **e**, The variations in the intensities of L_C and P_G , along with the interplanar spacing of (004) crystal plane, as external pressure increased. The upper schematic diagram illustrated that with increasing pressure, the coupling between host

and guest was strengthened and the decoupling process was suppressed. Therefore, the complex exhibited a red-shifted emission and the phosphorescence, P_G , was weakened. **f**, Rate constants of radiative and non-radiative decay from S_C state within PY–BrCzA complex under different pressure. **g**, Schematic diagram for different impact of static (left) and dynamic (right) coupling on RTP (P_G). In static coupling, the static complex was already formed in the ground state. Upon excitation to S_C and subsequent ISC to T_C , the static complex remained coupled and decayed back to the ground state, exhibiting L_C emission. In dynamic coupling, the dynamic complex was formed after excitation to the excited state. After undergoing ISC process to T_C , the dynamic complex dissociated and facilitated the transfer of excitons to T_G . Thus, the excitons decayed from T_G , exhibiting P_G emission.

On page 13 in the revised manuscript:

Beyond qualitative observations, we established a semi-empirical physical parameter to quantitatively characterize the dynamic coupling interaction (Supplementary Section VII). The activation energy barrier for the decoupling process (ΔG_{dc}^\ddagger) was calculated based on Marcus theory^{52,53} and systematically compared across various host–guest systems (Supplementary Figs. 96–99, Supplementary Tables 7 and 8). Our analysis revealed a strong inverse correlation ($R^2 = 0.99$) between ΔG_{dc}^\ddagger and RTP performance: higher energy barriers consistently corresponded to diminished phosphorescence lifetime (Supplementary Fig. 98). These quantitative results not only validated decoupling dynamics as a reliable regulatory mechanism for phosphorescence, but also shedding light on the design principles for host–guest RTP systems.

On page 20 in the revised manuscript:

52. Turro, N. J., Ramamurthy, V. & Scaiano, J. C. *Modern Molecular Photochemistry of Organic Molecules*. (University Science Books, New York, 2010).

53. Ma, L. et al. Triplet exciplex mediated multi-color ultra-long afterglow materials. *Angew. Chem. Int. Ed.* **64**, e202500847 (2025).

On page 23 in the revised manuscript:

We acknowledge Prof. Liangwei Ma and Prof. Xiang Ma for the helpful discussion.

On page 34 in the revised Supporting Information:

IV. Modulation of coupling strength through pressurization

.....

On page 38 in the revised Supporting Information:

Quantitative modulation of coupling strength: proximity effect

Fig. 4 demonstrates whether RTP is turned on depends on whether the coupling complex can undergo a decoupling process, enabling the excitons to transition from the T_C state to the triplet state

of guest, T_G . In the T_C state, electrons are delocalized across the entire complex, exhibiting CT characteristics, whereas in the T_G state, electrons are localized on the guest molecule, displaying locally excited (LE) characteristics. Thus, the decoupling process (i.e. T_C to T_G transition) inherently involves an electron exchange process to revert the CT state to the LE state. According to the Arrhenius equation (Eq. S1), the rate constant of this electron exchange process, k_{ex} , is governed by the activation energy barrier of decoupling process, ΔG_{dc}^\ddagger .

$$k_{ex} = k_0 \exp\left(-\frac{\Delta G_{dc}^\ddagger}{RT}\right) \quad \text{Eq. S1}$$

k_{ex} , rate constant of electron exchange during decoupling process; k_0 , pre-exponential factor, representing the maximum k_{ex} when ΔG_{dc}^\ddagger is 0 kcal/mol. ΔG_{dc}^\ddagger is the activation energy for decoupling process.

For a certain host–guest system, since the complex is formed through coupling between partially charged donor and acceptor moieties, whose relative motion is constrained by Coulombic interactions. In this context, the activation energy required for decoupling corresponds to the energy barrier needed to overcome the Coulombic potential. According to Coulomb’s law, this interaction energy (U) can be quantitatively described by Eq. S2.

$$U = \frac{1}{4\pi\epsilon_0\epsilon_r} \cdot \frac{q_1q_2}{r} \quad \text{Eq. S2}$$

ϵ_0 is the vacuum permittivity and ϵ_r is the relative permittivity. q_1 and q_2 represent the charges carried by the host and guest respectively. Since the host and guest are fixed in one identical system, their electron-donating/accepting capabilities remain constant, which means the quantities of partial charges they carried are fixed. Therefore, all these terms, ϵ_0 , ϵ_r , q_1 , and q_2 can be grouped into constant factors. r represents the intermolecular distance. Since the volume V of a solid scales with r^3 ($V \propto r^3$), and volume is inversely proportional to pressure p ($V \propto 1/p$), we can establish the relationship between U and p as shown in Eqs. S3–S4. Eq. S4 demonstrates that by varying external pressure, we can control the intermolecular distance and thereby modulate the activation energy for decoupling (ΔG_{dc}^\ddagger).

$$\frac{1}{r} \propto p^{1/3} \quad \text{Eq. S3}$$

$$\Delta G_{dc}^\ddagger = U = k_c \cdot p^{1/3} \quad \text{Eq. S4}$$

where k_c is a constant term determined by the intrinsic electronic properties of host and guest molecules, as well as the compressibility of the host crystals.

We take PY–BrCzA as an example and applied external pressure in the range of 0–10 GPa to modulate the activation energy by compressing intermolecular distances. Figure 4d shows its PXRD patterns under different pressure. All diffraction peaks shift toward higher angles with increasing pressure, indicating significant reduction in unit cell volume and interplanar spacing. Under 0–6 GPa external pressure, the main crystal diffraction peaks remain observable, demonstrating its crystal structure remain intact up to 6 GPa. At 10 GPa, the diffraction peaks weaken to near undetectable levels, indicating transition from crystalline to amorphous state. Thus, we can confirm that within the 0–6 GPa range, the proportional relationship between $1/r$ and $p^{1/3}$ in Eq. S3 is still valid, and consequently the correlation between ΔG_{dc}^\ddagger and $p^{1/3}$ in Eq. S4 remains effective.

Due to technical limitations in measuring pressure-dependent phosphorescence lifetimes, we utilize the phosphorescence intensity (I_p) as the performance metric. According to Eq. S1, we can derive an exponential relationship between I_p and ΔG_{dc}^\ddagger .

$$I_p \propto \exp(-\Delta G_{dc}^\ddagger)$$

Combining with Eq. S4, a negative linear correlation should exist between $\ln(I_p)$ and $p^{1/3}$.

$$\ln(I_p) \propto -p^{1/3}$$

As demonstrated by the experimental data, Supplementary Fig. 50 shows an exceptionally strong linear correlation ($R^2 = 0.99$) between $\ln(I_p)$ and $p^{1/3}$. Given the established relationship between $p^{1/3}$ and ΔG_{dc}^\ddagger in Eq. S4, this equivalently confirms a negative linear dependence of RTP intensity on the activation energy for decoupling (ΔG_{dc}^\ddagger). These results provide compelling experimental validation for our proposed mechanism: increasing external pressure elevates the decoupling activation energy through enhanced Coulombic interactions, thereby diminishing the RTP performance.

Supplementary Figure 50 | Relationship between phosphorescence intensity I_p and the activation energy for decoupling process, ΔG_{dc}^\ddagger , in PY-BrCzA system under different pressure.

UV-Vis absorption spectra

Supplementary Figure 51 demonstrates that the absorption spectra of PY-BrCzA crystals exhibit a progressive bathochromic shift in the onset position from 398 nm to 475 nm under increasing pressure. This observation clearly indicates that the electronic coupling between PY and BrCzA is enhanced through compressed intermolecular distance, thereby gradually lowering the energy gap of CT complex state. These spectral changes correlate well with the redshift of complex emission (L_c) observed in Fig. 4c. The parallel bathochromic trends in both absorption and emission spectra unambiguously provide evidence for the enhanced CT interactions between PY and BrCzA. Upon releasing the pressure back to ambient conditions, the absorption spectrum fully recovered to its original state, suggesting that this pressure-enhanced coupling process is reversible.

Supplementary Figure 51 | UV-Vis absorption spectra of PY-BrCzA under different pressure.

Time-resolved luminescence decay curves

To further investigate the evolution of complex emission (L_C), we measure its lifetime under varying pressure conditions. As the external pressure increases from 0 GPa to 10 GPa, the lifetime of L_C exhibits a significant decrease from 33 ns to 3 ns (Supplementary Fig. 52 and Supplementary Table 5). Detailed analysis of the radiative (k_r) and non-radiative (k_{nr}) decay rate constants (Fig. 4f) reveals a consistent trend: k_r progressively decreases while k_{nr} keeps increasing with pressure.

This phenomenon can be attributed to two primary factors: First, the enhanced CT character reduces the orbital overlap between HOMO and LUMO, leading to a more forbidden transition from S_C state (singlet state of coupling complex), thereby decreasing k_r . Second, the reduction in intermolecular distance between PY and BrCzA under pressure promotes molecular collisions and facilitates nonradiative energy transfer, resulting in the observed increase in k_{nr} . Thus, these lifetime measurements also strongly support our conclusion that increasing pressure enhances intermolecular coupling.

Supplementary Figure 52 | Lifetime curves of PY-BrCzA recorded at 500 nm under different pressure.

FT-IR spectra and Raman spectra

In situ FT-IR and Raman spectra under high pressure were also measured (Supplementary Figs. 53–55). The single-crystal structure of BrCzA reveals distinct C–H···Br interaction (Supplementary Fig. 53), which can be used as a sensitive indicator for monitoring the evolution of intermolecular interactions under pressure.

As illustrated in Supplementary Fig. 54, the majority of FT-IR absorption peaks exhibit blue shifts with increasing pressure, consistent with bond shortening induced by lattice compression. Some typical absorption bands are assigned. For example, the band at $\sim 1280\text{ cm}^{-1}$ corresponds to in-plane ring-stretching vibrations of the benzene moiety. Its progressive splitting and broadening under pressure directly reflect enhanced π – π stacking interactions. Characteristic stretching modes of C–H and C–Br bonds (annotated in Supplementary Fig. 53) display contrasting spectral trends: a continuous blue shift for $\nu(\text{C–H})$ from 2931 to 2983 cm^{-1} , versus a red shift for $\nu(\text{C–Br})$ from 734 to 723 cm^{-1} . This provides compelling evidence for the strengthening of C–H···Br interactions under high pressure.

Raman spectral evolution corroborates these findings (Supplementary Fig. 55). The C–H bending mode shows a monotonic blue shift, whereas the C–Br bending mode undergoes an initial red shift followed by a blue shift at higher pressures. This might be attributed to the intensified intermolecular interaction dominates at lower pressures, while the bond compression prevails at higher pressure.

Thus, the combined evidence from FT-IR spectra, Raman spectra, complemented by the PXRD analysis, conclusively confirms three key structural changes within PY–BrCzA under high pressure: lattice contraction, shortened intermolecular spacing and amplified intermolecular interactions. These phenomena provide corroborative support for the reinforcement of coupling interaction within the system.

Supplementary Figure 53 | C–H···Br interaction in BrCzA crystal.

Supplementary Figure 54 | FT-IR spectra of PY-BrCzA under different pressure.

Supplementary Figure 55 | Raman spectra of PY-BrCzA under different pressure.

On page 80 in the revised Supporting Information:

VII. Quantitative analysis of dynamic coupling

As we discussed in Supplementary Section IV, the activation energy ΔG_{dc}^\ddagger can be used as a semi-empirical parameter to quantify the dynamic coupling interaction and correlate it with RTP performance (Supplementary Fig. 96). According to Eq. S1, a higher activation energy corresponds to a slower electron exchange rate, thereby suppressing the decoupling process and weakening RTP emission from the T_G state, and *vice versa*.

$$k_{ex} = k_0 \exp\left(-\frac{\Delta G_{dc}^\ddagger}{RT}\right) \quad \text{Eq. S1}$$

Supplementary Figure 96 | Illustration of the relationship between ΔG_{dc}^\ddagger and RTP performance.

Below we will elaborate in detail on the application of this parameter to quantify dynamic coupling interactions in a series of host-guest systems investigated in our study. Specifically, we systematically analyzed the electronic effect in dynamic coupling process based on Marcus theory and compared ΔG_{dc}^\ddagger across different host-guest systems.

Given that the decoupling process intrinsically involves electron exchange, the relative electron-donating and electron-accepting capabilities of the host and guest components will influence the activation energy barrier and ultimately govern the decoupling propensity of the coupling complex. According to Marcus theory¹², ΔG_{dc}^\ddagger is associated with the thermodynamic free energy difference, ΔG_{dc}^0 , and reorganization energy, λ , during decoupling process (Eq. S5).

$$\Delta G_{dc}^\ddagger = \frac{(\Delta G_{dc}^0 + \lambda)^2}{4\lambda} \quad \text{Eq. S5}$$

ΔG_{dc}^0 , representing the energy difference between coupling complex and decoupled host/guest pairs after electron exchange, can be estimated by measuring the electrochemical potentials for the oxidations $E_{(D^+/D)}^0$ and reductions $E_{(A/A^-)}^0$ via cyclic voltammetry experiments¹², as per Eq. S6.

The organizational energy (λ) of the system, representing internal molecular reorganization from T_C state to T_G state, since external solvent reorganization is negligible in the crystal state, can be estimated using DFT¹³.

$$\Delta G_{dc}^0 \approx \mathcal{F}E_{(D^+/D)}^0 - \mathcal{F}E_{(A/A^-)}^0 - E_{D^*/A^*} \quad \text{Eq. S6}$$

E_{D^*/A^*} denotes the excitation energy of either the donor or acceptor, depending on which molecular entity is actually excited in the photophysical process.

Based on Eqs. S5–S6, we can determine the semi-empirical physical parameter ΔG_{dc}^\ddagger for each system by calculating the corresponding ΔG_{dc}^0 and λ values. Thus, we select four representative systems in our work with PY as guest and carbazole derivatives as hosts (PY–CzA, PY–BrCzA, PY–CzBP, and PY–BPCzA). To strengthen our dataset, we incorporate additional PY–BP system where benzophenone (BP) serves as the host. This is because BP represents the fundamental structural unit of CzBP/BPCzA, and PY–BP has been widely reported as an efficient RTP system^{14–16}. Their ΔG_{dc}^0 , λ and ΔG_{dc}^\ddagger values are listed in Supplementary Table 7. Cyclic voltammetry curves used for determining the electrochemical potentials are provided in Supplementary Fig. 97 and Supplementary Table 8.

Supplementary Figure 97 | Cyclic voltammetry curves of PY and host molecules in MeCN.

Correlation of ΔG_{dc}^\ddagger with RTP performance

As established in Section IV, the rate constant of electron exchange during decoupling process (k_{ex}) directly controls the population of excitons reaching T_G (Φ_T). For systems exhibiting comparable triplet decay rates ($k_P + k_{nr,P}$), the observed phosphorescence lifetime (τ_P) demonstrates a linear dependence on both Φ_T , and consequently, on k_{ex} .

$$\tau_P \propto \Phi_T \propto k_{ex} \quad \text{Eq. S7}$$

Therefore, we selected τ_P as the key performance metric for RTP, while ΔG_{dc}^\ddagger serves as our semi-empirical parameter to quantify the decoupling tendency in the dynamic coupling process.

Based on Eqs. S1 and S7, we can derive an exponential relationship between τ_p and ΔG_{dc}^\ddagger :

$$\tau_p \propto \exp(-\Delta G_{dc}^\ddagger)$$

Consequently, a negative linear correlation exists between $\ln(\tau_p)$ and ΔG_{dc}^\ddagger :

$$\ln(\tau_p) \propto -\Delta G_{dc}^\ddagger$$

Supplementary Fig. 98 reveals an excellent linear correlation ($R^2 = 0.99$) between $\ln(\tau_p)$ and ΔG_{dc}^\ddagger , which strongly supports the proposed mechanism of dynamic coupling-induced RTP. The observed negative slope reveals a systematic trend across the series from PY-CzBP to PY-BPCzA to PY-CzA to PY-BP: as the activation energy increases, the decoupling efficiency progressively decreases, resulting in corresponding reduction of RTP lifetimes, which is well consistent with our discussion in Supplementary Fig. 96. For the PY-BrCzA system, the heavy-atom effect (HAE) significantly increases the triplet decay rates ($k_p + k_{nr,p}$), which makes the proportional relationship between τ_p and k_{ex} in Eq. S7 invalid. Thus, this deviation can be excluded reasonably from the linear fitting analysis (Supplementary Fig. 99). Such an excellent linear correlation observed in Supplementary Fig. 98 strongly suggests that the phosphorescence performance in different host-guest systems can be quantitatively predicted by calculating the activation energy ΔG_{dc}^\ddagger .

Supplementary Figure 98 | Relationship between phosphorescence lifetime τ_p and the activation energy for decoupling process, ΔG_{dc}^\ddagger , for various host-guest systems.

Supplementary Figure 99 | (a) Deviation of PY–BrCzA system from fitted linear relationship due to heavy atom effect. (b) Experimental rate constants of radiative decay, k_p , and nonradiative decay, $k_{nr,p}$, from T_G state in these systems.

On page 94 in the revised Supporting Information:

Supplementary Table 5 | Photophysical properties of PY–BrCzA under different pressure.

External pressure (GPa)	τ (ns) ^a	k_r ($\times 10^5$ s ⁻¹)	k_{nr} ($\times 10^7$ s ⁻¹)
0.0	32.55	7.07	2.82
0.4	37.19	3.93	2.55
1.1	29.85	2.76	3.25
1.8	24.53	2.66	3.98
4.1	19.01	2.37	5.18
6.1	12.79	1.94	7.75
10.0	3.26	1.92	30.60

^a Lifetime was measured at 500 nm.

On page 95 in the revised Supporting Information:

Supplementary Table 7 | Thermodynamic parameters of host–guest systems during dynamic coupling process.

	PY–CzA	PY–BrCzA	PY–CzBP	PY–BPCzA	PY–BP
ΔG_{dc}^0 (kcal/mol)	-36.50	-38.28	-20.24	-34.55	-50.73
λ (kcal/mol)	7.40	10.67	15.27	16.81	8.81
ΔG_{dc}^\ddagger (kcal/mol)	28.60	17.85	0.40	4.69	49.88
τ_p (ms)	363	104	433	414	315

Supplementary Table 8 | The electrochemical potential for the oxidation and reduction of PY and host molecules.

	PY	CzA	BrCzA	CzBP	BPCzA	BP
$E_{(M^+/M)}^0$ (V)	1.20	1.10	0.80	1.27	1.49	2.10
$E_{(M/M^-)}^0$ (V)	-1.93	-2.20	-2.07	-1.55	-1.50	-1.65

On page 96 in the revised Supporting Information:

(12) Turro, N. J., Ramamurthy, V. & Scaiano, J. C. *Modern Molecular Photochemistry of Organic Molecules*. (University Science Books, New York, 2010).

(13) Ma, L. et al. Triplet exciplex mediated multi-color ultra-long afterglow materials. *Angew. Chem. Int. Ed.* **64**, e202500847 (2025).

(14) Xiao, F. et al. Guest-host doped strategy for constructing ultralong-lifetime near-infrared organic phosphorescence materials for bioimaging. *Nat. Commun.* **13**, 186 (2022).

(15) Qiu, W. et al. Achieving purely organic room-temperature phosphorescence mediated by a host-guest charge transfer state. *J. Phys. Chem. Lett.* **12**, 4600–4608 (2021).

(16) Yang, G. et al. Construction and application of large stokes-shift organic room temperature phosphorescence materials by intermolecular charge transfer. *J. Phys. Chem. Lett.* **14**, 6927–6934 (2023).

Considering that some new discussion, Supplementary Figures and references are added during the revision, the serial numbers of pages, Supplementary Figures, and references are also re-ordered in the revised manuscript and Supporting Information with double-check.

Reviewer #2 (Remarks to the Author):

Our comments are logically addressed. We suggest to accept the manuscript.

Response: We would like to express our gratitude for your thorough review and positive comments.

Reviewer #3 (Remarks to the Author):

The authors have well addressed my concerns. I think the quality of the manuscript is significantly improved. Thus, I recommend the acceptance of the manuscript by the journal as is.

Response: We appreciate your time and effort. Thank you for your positive comments.